# Robustness of statistical methods when measure is affected by ceiling and/or floor effect

**Matúš Šimkovic** *, **Birgit Träuble**

Universität zu Köln, Cologne, Germany

* matus.simkovic@uni-koeln.de

**Citation:** Šimkovic M, Träuble B (2019) Robustness of statistical methods when measure is affected by ceiling and/or floor effect. PLoS ONE 14(8): e0220889. https://doi.org/10.1371/journal.pone.0220889

**Data Availability Statement:** All relevant data are within the manuscript and its Supporting Information files. The code used to run the simulations and to generate the figures is available from https://github.com/simkovic/CFE.

## Abstract

### Goals and methods

A simulation study investigated how ceiling and floor effect (CFE) affect the performance of Welch's *t*-test, *F*-test, Mann-Whitney test, Kruskal-Wallis test, Scheirer-Ray-Hare-test, trimmed *t*-test, Bayesian *t*-test, and the "two one-sided tests" equivalence testing procedure. The effect of CFE on the estimate of group difference and on its confidence interval, and on Cohen's *d* and on its confidence interval was also evaluated. In addition, the parametric methods were applied to data transformed with log or logit function and the performance was evaluated. The notion of essential maximum from abstract measurement theory is used to formally define CFE and the principle of maximum entropy was used to derive probability distributions with essential maximum/minimum. These distributions allow the manipulation of the magnitude of CFE through a parameter. Beta, Gamma, Beta prime and Beta-binomial distributions were obtained in this way with the CFE parameter corresponding to the logarithm of the geometric mean. Wald distribution and ordered logistic regression were also included in the study due to their measure-theoretic connection to CFE, even though these models lack essential minimum/maximum. Performance in two-group, three-group and 2 × 2 factor design scenarios was investigated by fixing the group differences in terms of CFE parameter and by adjusting the base level of CFE.

### Results and conclusions

In general, bias and uncertainty increased with CFE. Most problematic were occasional instances of biased inference which became more certain and more biased as the magnitude of CFE increased. The bias affected the estimate of group difference, the estimate of Cohen's *d* and the decisions of the equivalence testing methods. Statistical methods worked best with transformed data, albeit this depended on the match between the choice of transformation and the type of CFE. Log transform worked well with Gamma and Beta prime distribution while logit transform worked well with Beta distribution. Rank-based tests showed best performance with discrete data, but it was demonstrated that even there a model derived with measurement-theoretic principles may show superior performance. Trimmed *t*-test showed poor performance. In the factor design, CFE prevented the detection

**Funding:** The authors received no specific funding for this work.

**Competing interests:** The authors have declared that no competing interests exist.

of main effects as well as the detection of interaction. Irrespective of CFE, *F*-test misidentified main effects and interactions on multiple occasions. Five different constellations of main effect and interactions were investigated for each probability distribution, and weaknesses of each statistical method were identified and reported. As part of the discussion, the use of generalized linear models based on abstract measurement theory is recommended to counter CFE. Furthermore, the necessity of measure validation/calibration studies to obtain the necessary knowledge of CFE to design and select an appropriate statistical tool, is stressed.

# 1 Introduction

In 2008, Schnall investigated how participants rate moral dilemmas after they have been presented with words related to the topic of cleanliness, as opposed to neutral words [1], [2]. The study reported that participants who were primed with the concept of cleanliness, found moral transgressions less bad than participants who weren't primed. [2] conducted a replication study using the same methods and materials. In contrast to [1, 2] found that the mean ratings of the two groups in their study did not differ. [3] pointed out that in [2] participants provided overall higher ratings than in the original study. [3] argued that the failure of the replication study to establish a difference between the two groups was due to a *ceiling effect*. Since substantial proportions of both groups provided maximum ratings, it was not possible to determine a difference in rating between these two groups. [4] provided their own analyses of ceiling effects in both, the original and the replication study and concluded that the ceiling effect can't account for the failure to replicate the original finding. Other researchers shared their analyses of the ceiling effects in these two studies on their websites and on social media. Here is a quick overview of the variety of the suggested analyses: [3] showed that the mean ratings in the replication study were significantly higher than those in the original study. Furthermore, she showed that the proportion of the most extreme ratings on the 10 point scale was significantly higher in the replication study than in the original study. [4] argued that rank-based Mann-Whitney test provides results that are identical to an analysis with Analysis of Variance (ANOVA). Furthermore, analyses without extreme values failed to reach significance as well. [5] did not find the above-mentioned analyses satisfactory and suggested an analysis with Tobit model, which showed a non-significant effect. [6] argued that Schnall's analyses do not support her conclusions. [7] investigated how ceiling effects would affect the power of a t-test. He used a graded response model to simulate data that were affected by ceiling, similar to those obtained in the replication study. The effect size was set to a value obtained in the original study. He found that, depending on the model parametrization, the power of a t-test in the simulated replication study ranges from 70 to 84% which should be sufficient to detect the effect. [8] performed Bayes Factor analysis and compared the quantiles. Both analyses suggested an absence of an effect in the replication study.

In our opinion, this discussion about the presence and impact of ceilings effects illustrates how relevant, yet elusive, the concept of ceiling effect is. Apparently, the only point regarding ceiling effects, on which all parties agreed, is that the application of parametric analyses such as ANOVA or t-test is problematic in the presence of a ceiling effect. Yet the authors disagreed on how to demonstrate and measure the impact of ceiling effect, which makes the default application of ANOVA problematic per se. Motivated by these concerns, the current work presents a computer simulation study that investigates how various methods of statistical

inference perform when the measurements are affected by ceiling and/or floor effect (CFE). The main focus is on the performance of the textbook methods: Welch's version of t-test, ANOVA and rank-based tests. In addition, the performance of potential candidate methods, some of which were already encountered in the discussion of the study by [1], is investigated. The hallmark of the current work is the theoretical elaboration of the concept of CFE with the help of formal measurement theory [9]. This theoretical embedding provides a backbone for the simulations, and, as we further point out, the lack of such theoretical embedding may be one of the reasons why the number and scope of simulation studies of CFE up until now is limited.

Measurement-theoretic definitions of CFE are discussed in section 1.1. Section 1.2 reviews the literature on the robustness of the textbook methods. Only few robustness studies do explicitly consider the CFE. However, numerous studies investigate other factors, which may combine to create CFE. These studies, in particular, are considered in section 1.2. Taking stock of the material presented in the preceding sections, sections 1.3.1 and 1.3.2 justify the choice of statistical methods and the choice of the data-generating mechanism utilized by the simulations. While section 2 provides additional details on the methods and procedures, the description provided in sections 1.3.1 and 1.3.2 should be sufficient to follow the results presentation and a reader interested in the results may directly skip to section 3 and 4.

## 1.1 Formal definition of CFE

Consider first some informal notions of CFE. The dictionary of statistical terms by [10] provides the following entry on CFE. "Ceiling effect: occurs when scores on a variable are approaching the maximum they can be. Thus, there may be bunching of values close to the upper point. The introduction of a new variable cannot do a great deal to elevate the scores any further since they are virtually as high as they can go." The dictionary entry in [11] (see also [12]) says: "Ceiling effect: A term used to describe what happens when many subjects in a study have scores on a variable that are at or near the possible upper limit ('ceiling'). Such an effect may cause problems for some types of analysis because it reduces the possible amount of variation in the variable."

We identify two crucial aspects of CFE in these quotes. First, CFE causes a "bunching" of measured variables, such that the measure becomes insensitive to changes in the latent variable that it is supposed to measure. Second, CFE does not only affect the expected change in the measured variable but its other distributional properties as well which may in turn affect the performance of some statistical methods. [11] mentions the variability, which one may interpret as the variance of the measured variable. [13] hypothesized that skew is the crucial property that characterizes CFE. Importantly, the informal descriptions lack precise rationale and risk excluding less obvious and intuitive phenomena from the definition of CFE. In section 1.1.1 we show that formal measurement theory allows us to make these (and many others) informal notions precise. Historically, the research in measurement theory has been concerned with deterministic variables and despite multiple attempts a principled extension to random variables was not achieved. In section 1.1.3 we review the derivation of maximum entropy distributions, which provides an extension of measurement theory to random variables and in particular allow us to derive distributions, that can be used to simulate CFE and to manipulate its magnitude.

**1.1.1 Measurement theory.** A function, say $\phi$ from $A$ to a subset of $\mathbb{R}$, describes the assignment of numbers to empirical objects or events. $\phi$ is, in the context of measurement theory, referred to as *scale*. It is crucial that $\phi$ is chosen such that the numerical values retain the relations and properties of the empirical objects in $A$ (i.e. $\phi$ is a homomorphism, see section

1.2.2 in [9]). For instance, if the empirical objects are ordered, such that $a \preceq b$ for some $a, b \in A$, then it is desirable that $\phi$ satisfies $a \preceq b$ if and only if (iff) $\phi(a) \leq \phi(b)$. Measurement theory describes various scale types and the properties of the empirical events necessary and sufficient to construct the respective scale type. In addition, given a set of properties of empirical objects, multiple choices of $\phi$ may be possible and measurement theory delineates the set of such permissible functions. A scale that preserves the order, i.e. $a \preceq b$ iff $\phi(a) \leq \phi(b)$ is referred to as *ordinal* scale. Given that $\phi$ is an ordinal scale, then $\phi'(a) = f(\phi(a))$ is also an ordinal scale for all $a \in A$ and for a strictly increasing $f$ (ibid. p.15). Note that the set of possible scales is described as a set of possible transformations $f$ of some valid scale $\phi$. Other notable instances are *ratio* scale and *interval* scale. In addition to order, a ratio scale preserves a concatenation operation $\circ$ such that $\phi(a \circ b) = \phi(a) + \phi(b)$. A ratio scale is specified up to a choice of unit i.e. $\phi'(a) = \alpha\phi(a)$, with $\alpha > 0$. The required structure of empirical events is called *extensive structure* (ibid. chapter 3). In some situations it is not possible to take direct measurements of the empirical objects of interest, however one may measure pairwise differences or intervals between the empirical objects, say $a \ominus b$ or $c \ominus d$. Then, one may construct an interval scale given that $a \ominus b \preceq c \ominus d$ if $\phi(a \ominus b) \leq \phi(c \ominus d)$ and $\phi(a \ominus c) = \phi(a \ominus b) + \phi(b \ominus c)$ for all $a, b, c, d \in A$ (ibid, p. 147). The set of permissible transformations is given by $\phi'(a) = \alpha\phi(a) + \beta$ with $\beta \in \mathbb{R}$ and $\alpha > 0$. The corresponding structure is labelled *difference structure* (ibid. chapter 4).

Consider concatenation again. The concatenation operation in length measurement can be performed by placing two rods sequentially. In weight measurement the concatenation may be performed by placing two objects on the pan of a balance scale. As [9] (chap. 3.14) point out, finding and justifying a concatenation operation in social sciences often poses difficulties. Furthermore, it is not necessary to map concatenation to addition. For instance, taking $\psi = \exp(\phi)$, with interval scale $\phi$, will translate addition on $\mathbb{R}$ to multiplication on $\mathbb{R}^+$. More generally, any strictly monotonous function $f$ may be used to obtain a valid numerical representation $\psi(a) = f^{-1}(\phi(a))$ with a (possibly non-additive) concatenation formula $\psi(a \circ b) = f^{-1}(f(\psi(a)) + f(\psi(b)))$. [9] (chap. 3.7.1) make use of this fact when considering measurement of relativistic velocity (also referred to as rapidity). Relativistic velocity is of interest in the present work because it is bounded—it can't exceed the velocity of light. The upper bound poses difficulties for the additive numerical representation since an extensive structure assumes positivity of addition, i.e. $a \prec a \circ b$ for all $a, b \in A$ (axiom 5 in definition 1 on p.73 ibid.). However, if $z$ is the velocity of light, we have $z \sim z \circ a$, which violates positivity. [9] resolve this issue by mapping velocity from a bounded range to an unbounded range, performing addition there and mapping the result back to the bounded range. Formally, concatenation is given by ([9] chapter 3, theorem 6)

$$f_u\left(\frac{\phi(a \circ b)}{\phi(z)}\right) = f_u\left(\frac{\phi(a)}{\phi(z)}\right) + f_u\left(\frac{\phi(b)}{\phi(z)}\right) \tag{1}$$

where $f_u$ is a strictly increasing function from $[0, 1]$ to $\mathbb{R}^+$ that is unique up to a positive multiplicative constant. As [9] point out, taking transformation $f_u = \tanh^{-1}$ results in the velocity-addition formula of the relativistic physics. However, this choice is arbitrary and Eq 1 provides us with the general result, which we will use in the current work. [9] call an element which satisfies $z_u \sim z_u \circ a$ (for all $a \in A$) an *essential maximum*. [9] further show that given an extensive structure with an essential maximum, there always exists a strictly increasing function $f_u$ such that Eq 1 is satisfied.

Next, we consider several straightforward extensions of the result by [9]. First, we wish to introduce extensive structures with an essential minimum $z_l$. Note that even though velocity has a lower bound at zero, this lower bound is not an essential minimum, because the

concatenation is positive and a repeated concatenation results in increasing numerical values. As a consequence, essential maximum or essential minimum is a property related to a concatenation operation rather than a property of the numerical range of a scale. If we distinguish between $z_u$ and $z_l$ we need to distinguish between $\circ_u$ and $\circ_l$ and in turn between $f_u$ and $f_l$. Of course, this does not preclude the possibility that in some particular application it may be true that $f_u = f_l$.

Consider a modification of Eq 1 to describe an essential minimum. Qualitatively, an essential minimum manifests, similar to an essential maximum, the property $z \sim z \circ_l a$. In this case however $\circ_l$ is a negative operation in the sense that $a \succ a \circ_l b$ for all $a, b \in A$. The results then are analogous to those of the velocity derivation. The main difference is that the scale $\phi_l$ maps to $[\phi_l(z_l), 0]$ rather than to $[0, \phi_u(z_u)]$. However, both expressions $\phi_l(a)/\phi_l(z_l)$ and $\phi_u(a)/\phi_u(z_u)$ translate into the range $[0, 1]$ and hence, the only modification of Eq 1 is to add subscripts:

$$f_l\left(\frac{\phi_l(a \circ_l b)}{\phi_l(z_l)}\right) = f_l\left(\frac{\phi_l(a)}{\phi_l(z_l)}\right) + f_l\left(\frac{\phi_l(b)}{\phi_l(z_l)}\right). \tag{2}$$

Thus, if a structure has an essential minimum, then a strictly increasing function $f_l$ exists such that Eq 2 is satisfied.

Second, we wish to extend Eq 1 to situations in which the minimal element $z_l$ (irrespective of whether it is essential minimum or not) is non-zero. We do so by first translating the measured values from range $[\phi(z_l), \phi(z_u)]$ to $[0, \phi(z_u) - \phi(z_l)]$ and then to the domain of $f_u$ i.e. $[0, 1]$:

$$f_u\left(\frac{\phi_u(a \circ_u b) - \phi_u(z_l)}{\phi_u(z_u) - \phi_u(z_l)}\right) = f_u\left(\frac{\phi_u(a) - \phi_u(z_l)}{\phi_u(z_u) - \phi_u(z_l)}\right) + f_u\left(\frac{\phi_u(b) - \phi_u(z_l)}{\phi_u(z_u) - \phi_u(z_l)}\right). \tag{3}$$

Third, similar to step two, we modify Eq 2 to apply in situations with a non-zero maximal element $z_u$

$$f_l\left(\frac{\phi_l(a \circ_l b) - \phi_l(z_u)}{\phi_l(z_l) - \phi_l(z_u)}\right) = f_l\left(\frac{\phi_l(a) - \phi_l(z_u)}{\phi_l(z_l) - \phi_l(z_u)}\right) + f_l\left(\frac{\phi_l(b) - \phi_l(z_u)}{\phi_l(z_l) - \phi_l(z_u)}\right). \tag{4}$$

Above, we distinguished between a scale with an essential minimum $\phi_l$ and a scale with an essential maximum $\phi$, which was in Eq 3 labelled more accurately as $\phi_u$. This distinction was necessary, because the two scales in Eqs 3 and 4 map to different number ranges. However, and this is the fourth extension, we need to consider a single scale which has both, an essential minimum and an essential maximum. To do so, consider a scale $\phi$ with range $[\phi(z_l), \phi(z_u)]$. Then both Eqs 3 and 4 apply. We just change the labels: $\phi_l = \phi_u = \phi$.

Fifth, a further simplification can be achieved by assuming that essential minima and essential maxima affect concatenation in an identical manner i.e. $f = \alpha_u f_u = \alpha_l f_l = f$ for some positive constants $\alpha_l$ and $\alpha_u$. Eqs 3 and 4 simplify respectively to:

$$f(g[\phi(a \circ_u b)]) = f(g[\phi(a)]) + f(g[\phi(b)]) \tag{5}$$

$$f(1 - g[\phi(a \circ_l b)]) = f(1 - g[\phi(a)]) + f(1 - g[\phi(b)]) \tag{6}$$

for all $a, b \in A$. To simplify the notation, we introduced the function $g[x] = \frac{x - \phi(z_l)}{\phi(z_u) - \phi(z_l)}$. This notation highlights that in terms of measurement $\phi$, the operations $\circ_l$ and $\circ_u$ are symmetric around the line $g[x] = 0.5$. To illustrate this with an example, consider the popular choice $f_l(x) = f_u(x) = f(x) = -\log(1 - x)$ with $x$ restricted to $[0, 1]$. This is a strictly increasing function to $\mathbb{R}^+$ and hence provides a valid choice. The left panel in Fig 1 shows $f(1 - g[\phi(a)]) = -\log(g[\phi(a)])$

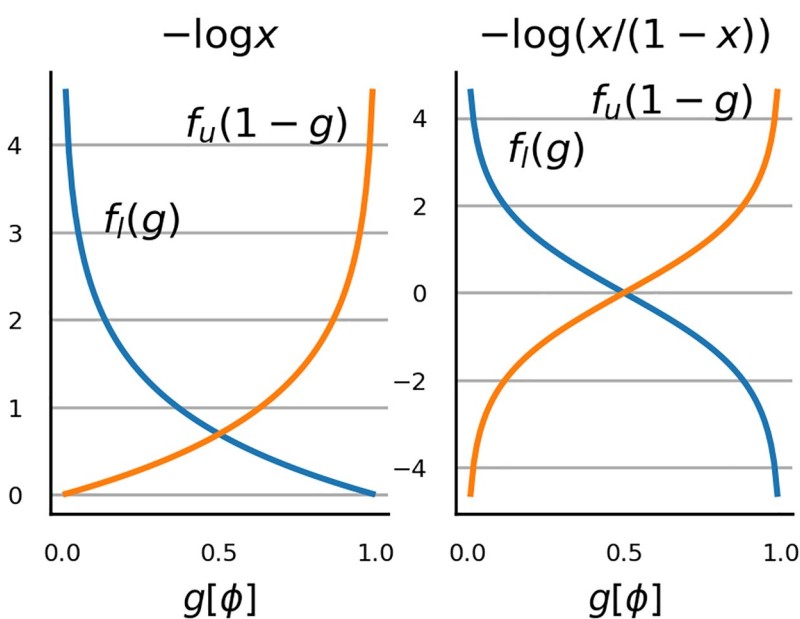

**Fig 1. Examples of $f_u$ and $f_l$.** The panels show functions $f_l(g[\phi])$ and $f_u(1 - g[\phi])$. In the left panel $f_l(g[\phi]) = f_u(g[\phi]) = \log g[\phi]$ while in the right panel $f_l(g[\phi]) = f_u(g[\phi]) = \log(g[\phi]/(1 - g[\phi]))$.

and $f(g[\phi(a)]) = -\log(1 - g[\phi(a)])$ as a function of $g[\phi(a)] \in [0, 1]$. As noted, the two curves manifest symmetry around $g[\phi(a)] = 0.5$

Above, we assumed that $\phi$ is a ratio scale. As a final modification we consider the case when $\phi$ is an interval scale. The result for an interval scale and the corresponding difference structure is provided in chapter 4.4.2 in [9]. The result is identical to Eq 1 except that $f$ is a function to $\mathbb{R}$ rather than to $\mathbb{R}^+$ and that $f$ is unique up to a linear transformation. Hence, if $\phi$ is an interval scale and $\circ_l = \circ_u$ then $f = \alpha_u f_u + \beta_u = \alpha_l f_l + \beta_l$ ($\alpha > 0$ and $\beta \in \mathbb{R}$). Again, to provide an example of an interval scale with an essential maximum and an essential minimum, consider a case with logit function $f_l(x) = f_u(x) = f(x) = \log(x/(1 - x))$. The right panel in figure shows $f(1 - g[\phi(a)])$ and $f(g[\phi(a)])$ as a function of $g[\phi(a)] \in [0, 1]$. A logit function is a strictly increasing function from $[0, 1]$ to $\mathbb{R}$ and is hence valid model for difference structure with essential maximum and with essential minimum. The set of permissible transformations of $f$ is given by $f(x) = \alpha \log(x/(1 - x)) + \beta$.

**1.1.2 Structure with Tobit maximum.** The measurement structures discussed so far have the notable property that it's not possible to obtain the maximal element by concatenation of two non-maximal elements. [9] note that "we do not know of any empirical structure in which the concatenation of two such elements is an essential maximum". [9] (theorem 7 on p. 95-96) nevertheless provide the results for such a case, which we refer to as extensive structure with *Tobit maximum*. This measurement structure is implicit in the popular Tobit model which is sometimes discussed in connection with CFE and hence, we briefly present it. The scale $\phi$ must satisfy order monotonicity and monotonicity of concatenation when the concatenation result is not equal to the Tobit maximum. In the remaining case, i.e. when $z_u \sim a \circ b$, the concatenation is represented numerically as $\phi(z_u) = \inf(\phi(a) + \phi(b))$ where inf is the infimum over all $a, b \in A$ which satisfy $z_u \sim a \circ b$. The scale $\phi$ is unique up to a multiplication by a positive constant. Extensions to extensive structures with Tobit minimum and to difference structures with Tobit maximum and/or minimum are straightforward and follow the rationale presented in the previous section.

**1.1.3 Random variables with CFE.** The extensive and difference structures with essential minimum/maximum introduce a crucial aspect of CFE that was exemplified by the dictionary entry on the ceiling effect by [10]. Formally, we may interpret the "introduction of a new variable" that elevates the previous level of the variable as a concatenation with some other element. Then we may look at the difference between the new level $f^{-1}(f(x) + f(h))$ and the old level $x$. Notably, we get $\lim_{x \to 1}|f^{-1}(f(x) + f(h)) - x| = 0$ (for $h \neq 0$), which may be seen as a formal notion of "bunching". Crucially, measurement theory suggests that the concept of a boundary effect implicitly assumes the existence of a concatenation operation which has a non-additive numerical representation.

The measurement-theoretic account fits well with the description of CFE by [10]. It misses however the other highlighted aspects of CFE: the distributional properties of the measured variable such as the variance reduction or the increased skew. To formally approach the concept of reduced variation and the influence of CFE on the distribution of the measured values more generally, we need to introduce the concept of random variables (RVs). Recall, that scale $\phi$ maps from the set of empirical events to some interval. Instead, we consider $\phi$ to map from empirical events to a set of RVs over the same interval. We are not aware of work that explores such a probabilistic formulation of measurement theory. Nor do we wish to explore such an approach in detail in the current work. Our plan is to point out that the above-listed results from measurement theory along with few straight-forward assumptions about the probabilistic representation, allow us to derive most widely used probability distributions. Crucially, the derivation determines which parameters and under what transformation, represent the concatenation operation. This in turn allows us to specify and justify the choice of data generators used in the simulations and the choice of the metrics used to evaluate the performance on the simulated data.

When the scale maps to a set of RVs, constraints, such us $\phi(a \circ b) = \phi(a) + \phi(b)$, or $a \preceq b$ iff $\phi(a) \leq \phi(b)$, are in general not sufficient to determine the distribution of $\phi$. The first step is to formulate the constraints in terms of summaries of RVs. We choose the expected value $E[X]$ as data summary. The expected value is linear in the sense that $E[aX + b] = aE[X] + b$ and also $E[X] + E[Y] = E[X + Y]$ (where $X, Y$ are independent RVs and $a, b$ are constants). Due to the linearity property, we view expected value as a data summary that is applicable to scales, which represent concatenation by addition. See chapter 2 in [14] and chapter 22 in [15] for similar views on the role of expectation in additive numerical representations.

As a consequence, we modify the constraint $a \preceq b$ iff $\phi(a) \leq \phi(b)$ to $a \preceq b$ iff $E[\phi(a)] \leq E[\phi(b)]$. We modify the constraint $\phi(a \circ b) = \phi(a) + \phi(b)$ to $E[\phi(a \circ b)] = E[\phi(a)] + E[\phi(b)]$. Effectively, the above constraints state that $\phi$ is a parametric distribution with a parameter equal to $E[\phi(a)]$ for each $a \in A$ and in which the parameter satisfies monotonicity, additivity, or some additional property required by the structure.

Above, we saw that in the presence of CFE, concatenation can't be represented by addition. Instead, we apply the expectation to the values transformed with $f$ which supports addition. For instance Eq 3 translates into

$$
\begin{aligned}
E\left[f_u\left(\frac{\phi_u(a \circ_u b) - \phi_u(z_l)}{\phi_u(z_u) - \phi_u(z_l)}\right)\right] &= E\left[f_u\left(\frac{\phi_u(a) - \phi_u(z_l)}{\phi_u(z_u) - \phi_u(z_l)}\right)\right] \\
&+ E\left[f_u\left(\frac{\phi_u(b) - \phi_u(z_l)}{\phi_u(z_u) - \phi_u(z_l)}\right)\right].
\end{aligned}
\tag{7}
$$

Thus, we require that the distribution is parametrized by $c_u = E[(f_u(g[\phi(a)])]$ and/or $c_l = E[(f_l(1 - g[\phi(a)])]$ depending on whether the structure has an essential maximum, an essential minimum or both.

Consider again the dictionary descriptions of CFE. One may interpret these in terms of random variables as follows. With the repeated concatenation, the expected value of the measured values approaches the boundary. Furthermore, as it approaches the boundary, a concatenation of the equivalent object/event results in an increasingly smaller adjustment to the expected value. Finally, [11] stated that the variability decreases as the values approach boundary, which one may interpret as that the variance of the random variable approaches zero upon repeated concatenation.

Consider a random variable $Y(c_u) \in [y_l, y_u]$ with a ceiling effect at $y_u$ and parameter $c_u = E[(f_u(g[\phi(a)])] \in \mathcal{R}$. We may investigate whether the stated requirements are satisfied by checking whether the following formal conditions of a ceiling effect are true:

1. $\lim_{c_u \to \infty} E[Y(c_u)] = y_u$

2. $\lim_{c_u \to \infty} E[Y(c_u + x)] - E[Y(c_u)] = 0$ for every $x > 0$

3. $\lim_{c_u \to \infty} Var[Y(c_u)] = 0$

4. $\lim_{c_u \to \infty} Skew[Y(c_u)] = \infty$.

Instead of condition 3 and 4, one may alternatively consider whether $Var[Y(c_u)]$ and $Skew[Y(c_u)]$ are respectively increasing and decreasing functions of $c_u$. Eq 7 implies the existence of series of random variables $Y(c_u)$ that converge to $\phi(z_u)$ as $c_u \to \infty$. By dominated convergence theorem (chapter 9.2 in [16]) then the second condition holds but instead of the first condition we obtain $\lim_{c_l \to -\infty} E[Y(c_l)] = E[\phi(z_l)]$. By assuming $E[\phi(z_l)] = y_l$ one additionally obtains both, the first and the third condition. Indeed, the third condition is a direct consequence of the first condition. Analogous results follow for a variable with a floor effect at $y_l$ with the limiting process $c_u \to -\infty$. To conclude, the second condition follows immediately from the measurement-theoretic considerations, however to determine the remaining conditions one has to consider specific distributions of $Y$ which we do next.

**1.1.4 Maximum entropy distributions with CFE.** In this section we present an approach that allows us to derive probability distribution $Y(c_l, c_u)$ given functions $f_l$ and $f_u$. We adapt the principle of maximum entropy (POME, [17–19]) to obtain the probability distributions with the desired parametrization. According to POME, if nothing else is known about the distribution of $Y$ except a set of $N$ constraints of the form $c_i = E[g_i(Y)]$ and that the domain of $Y$ is $[y_l, y_u]$ (and that it is a probability distribution, i.e. $c_0 = \int_{y_l}^{y_u} p(Y = y) dy = 1$), then one should select a distribution that maximizes the entropy of the distribution subject to the stated constraints. Mathematically, this is achieved with the help of the calculus of variations ([20] chapter 12). The POME derivation results in a distribution with $N$ parameters and the derivation fails if the constraints are inconsistent. The procedure is similar, but somewhat more general compared to the alternative method of deriving a parametric distribution that is member of the exponential family with the help of constraints ([for applications of this method see for instance [21]). POME allows to derive distributions that are not part of the exponential family. To mention some general examples, the uniform distribution is the maximum entropy distribution of RV on closed interval without any additional constraints. Normal distribution is the maximum entropy distribution of RV $Y$ on real line with constraints $c_m = E[Y]$ and $c_v = E[(Y - c_1)^2]$ ([19] section 3.1.1).

Table 1 provides an overview of maximum entropy distributions found in the POME literature that are derived from constraints posed by structures with essential minimum and/or essential maximum and thus relevant in the current context. For more details on the derivation of the listed maximum entropy distributions see [19] and [22]. We make the following

**Table 1. Maximum entropy distributions derived from constraints posed by structures with essential minimum and/or essential maximum.**

| distribution | range | constraint on $c_u$ and $c_l$ | additional constraint | $p(Y = y)$ | parameters |
|---|---|---|---|---|---|
| Beta | $Y \in [0, 1]$ | $c_u = E[\log(1-Y)]$ | | $B(a,b)^{-1} y^a(1-y)^b$ | $c_u = \psi(a) - \psi(a+b)$ |
| | | $c_l = E[\log(Y)]$ | | | $c_l = \psi(b) - \psi(a+b)$ |
| Logit-normal | $Y \in [0, 1]$ | $c_u = E[\log(Y/(1-Y))]$ | $c_v = Var[\log(Y/(1-Y))]$ | $(c_v(y-y^2)\sqrt{2\pi})^{-1} \times$ $\exp\left(-(\log[y/(1-y)] - c_u)^2/(\sqrt{2}c_v)^{-2}\right)$ | $c_u = c_l$ |
| | | $c_l = E[\log(Y/(1-Y))]$ | | | |
| Truncated Gamma | $Y \in [0, 1]$ | $c_l = E[\log(Y)]$ | $c_m = E[Y]$ | $\frac{1}{D(a,b)} \exp(-ay)y^b$ | $c_m = \int_0^1 yp(Y=y)dy$ |
| | | | | $D(a,b) = \int_0^1 \exp(-ay)y^b dy$ | $c_l = \int_0^1 \log y p(Y=y)dy$ |
| Generalized Gamma | $Y \in [0, \infty]$ | $c_l = E[\log(Y)]$ | $c_m = E[Y^{c_n}]\ c_n > 0$ | | $c_l = \log b + \psi(a)/c_n$ |
| | | | | $\frac{c_n}{y\Gamma(a)}\left(\frac{y}{b}\right)^{c_n a} \exp\left(-\left(\frac{y}{b}\right)^{c_n}\right)$ | $c_m = b^{c_n}a$ |
| Beta Prime | $Y \in [0, \infty]$ | $c_l = E[\log(Y)]$ | $c_n = E[\log(1+Y)]$ | | $c_l = \psi(a) - \psi(b)$ |
| | | | | $B(a,b)^{-1} y^{a-1}(1+y)^{-a-b}$ | $c_n = \psi(a+b) - \psi(b)$ |
| Log-normal | $Y \in [0, \infty]$ | $c_l = E[\log(Y)]$ | $c_v = Var[\log(Y)]$ | $(c_v y\sqrt{2\pi})^{-1} \cdot \exp\left(-\frac{(\log y - c_l)^2}{2c_v^2}\right)$ | |
| Generalized Geometric | $Y \in \{0, 1, \ldots, \infty\}$ | $c_l = E[\log(Y)]$ | $c_m = E[Y]$ | $D(a,b)^{-1} a^y y^b$ | $c_l = \exp\frac{\partial D}{\partial b}$ |
| | | | | $D(a,b) = \sum_{y=0}^\infty a^y y^b$ | $c_m = \frac{D(a,b+1)}{D(a,b)}$ |
| Discrete Beta | $Y \in \{0, 1, \ldots, c_n\}$ | $c_u = E[\log(1-Y)]$ | | $D(a,b)^{-1} y^a(1-y)^b$ | $c_u = \exp\frac{\partial D}{\partial a}$ |
| | | $c_l = E[\log(Y)]$ | | $D(a,b) = \sum_{y=1}^\infty y^a(1-y)^b$ | $c_l = \exp\frac{\partial D}{\partial b}$ |
| Beta-binomial | $Y \in \{0, 1, \ldots, c_n\}$ | $c_u = E[\log(1-Q)]$ | $E[Y] = q$ | $\binom{c_n}{y} B(y+a, c_n-y+b)/B(a,b)$ | $c_u = \psi(a) - \psi(a+b)$ |
| | | $c_l = E[\log(Q)]$ | $q \sim Beta(a, b)$ | | $c_l = \psi(b) - \psi(a+b)$ |

PDF—probability density function. $B$ is beta function. $\psi$ is Digamma function.

observations. First, the popular choice $f_u = f_l = -\log(1-Y)$ translates into constraints $c_u = E[\log(1-Y)]$ and $c_l = E[\log Y]$, which correspond to the logarithm of geometric mean of $Y$ and of $1 - Y$ respectively. The sole exception in the table is the Logit-normal distribution which uses $f_u = f_l = \log(Y/(1-Y))$ to model CFE.

Second, the maximum entropy distributions include large portion of the most popular distributions on $[0, 1]$ and $[0, \infty]$. Exponential, Weibull, Gamma, F, Log-logistic or Power function distribution are included by Generalized Gamma family, by Beta Prime distribution or both [23]. A generalized version of the Beta prime distribution can be obtained with a constraint $c_m = E[\log(1 + Y^{c_n})]$. [22] went further and showed that a maximum entropy distribution with constraints $c_l = E[\log(Y)]$ and $c_m = E[\log(1 + c_p Y^{c_n})/c_p]$ includes the Generalized Gamma family (for $c_p \to 0$) and the generalized Beta Prime family ($c_p = 1$). While [22] don't consider the case $c_p = -1$, it is straightforward to see that assuming $Y^{c_n} \in [0, 1]$ results in a generalized form of Beta distribution. The resulting distributions and the relations between them are discussed in more detail by [23].

Third, the parameters $c_v$ may be interpreted as variance parameters, or more generally as constraints that introduce a distance metric. Notably, the variance is formulated over $f(Y)$. Thus while $c_l$ can be respectively interpreted as expected log-odds or logarithm of geometric mean, $c_v$ can be interpreted as log-odds variance or geometric variance. Introduction of these parameters is consistent with the measurement theoretic framework, except that the constraints are expressed in terms of variance $c_v = Var[f(Y)]$ rather than in terms of expectation $c_l = E[f(Y)]$.

Fourth, the interpretation of the $c_m$ parameters is interesting as well. One possibility is to view $c_m$ as an additional constraint unrelated to CFE. As [24] discuss in the case of gamma

distribution, the parameter $c_l$ controls the generation of small values while $c_m$ controls the generation of large values. This interpretation is similar to the interpretation of $c_l$ and $c_u$ of beta distribution even though there is no essential maximum in the former case as opposed to the latter case.

Fifth, one may illustrate the similarity between $c_u$ of Beta and $c_m$ of Gamma distribution in one additional way. Consider a generalization of Beta distribution with a scaling parameter $b = \phi(z_u)$ such that $Y \in [0, b]$. As detailed in [23] Gamma and some other distributions can be obtained from a generalized Beta by constructing the limit $b \to \infty$. The parameter $c_u$ of Beta translates into $c_m$ of Gamma.

Sixth, it's possible for two notationally distinct constraints to result in the same probability distribution, albeit with different parametrization. For instance, constraints $c_l = \mathrm{E}[\log{(Y_n^c)}] = c_n \mathrm{E}[\log{(Y)}]$ and $c_m = \mathrm{E}[Y^{c_n}]$ imply a Generalized Gamma distribution with a somewhat different parametrization than that of the Generalized Gamma distribution listed in the table: $c_l = c_n \log b + \psi(a)$.

Seventh, recall that $f_l$ and $f_u$ are defined up to a scaling constant (extensive structure) or up to linear transformation (difference structure). As a consequence $c_l$ and $c_u$ are known up to a scale or up to a linear transformation, i.e. $c_l = \beta_l + \alpha_l \mathrm{E}[f_l(Y)]$. In similar manner, one may modify the constraints $c_u$ or even $c_m$ so that the set of permissible transformations of $f$ is explicit. $\alpha$ and $\beta$ are not identifiable in addition to $c_l$ and their introduction does not affect the derivation of maximum entropy distribution. Nevertheless, it may be possible to parametrize the distribution with $\alpha$ and/or $\beta$ instead of some nuisance parameter. Consider the Generalized Gamma distribution with constraints $c_l = \mathrm{E}[\log{(Y^{c_n})}]$ and $c_m = \mathrm{E}[Y^{c_n}]$. Note that we may introduce transform $c_l = c_n \mathrm{E}[\log(Y)]$ so that $c_n$ can be interpreted as the scale/unit of $c_l$. Finally, one may sacrifice parameter $c_m$ and introduce shift parameter, say $c_\beta$ such that $c_l = c_\beta + c_n \mathrm{E}[\log(Y)]$. From the formula for $c_l$ from Table 1 it follows that $c_\beta = c_n \log b = \log c_m - \log a$.

Eighth, as illustrated with the case of truncated gamma distribution, it is straightforward to introduce a maximum of the distribution while maintaining the floor effect. Note though that the maximum thus introduced is not an essential maximum and the process of truncation can't be used to introduce CFE. This can be perhaps best seen on the case of truncated normal distribution with range $[0, \infty]$ which does not satisfy the CFE conditions listed in the previous section.

Finally, it is straightforward to apply the above ideas to discrete measurement. While the values of $Y$ are discrete, the values $\mathrm{E}[f(Y)]$, that are part of the constraints, are continuous. Maximum entropy derivation provides analogous results to continuous distributions. Constraints $c_l = \mathrm{E}[\log Y]$ and $c_m = \mathrm{E}[Y]$ with $Y \in \{0, 1, \ldots\}$ lead to generalized geometric distribution ([19] section 2.1d). Constraints $c_l = \mathrm{E}[\log Y]$ and $c_m = \mathrm{E}[\log(1 - Y)]$ result in a discrete version of Beta distribution with $Y \in \{0, 1/n, 2/n, \ldots, (n - 1)/n, 1\}$ and $n \in \mathbb{N}$. Discrete Beta distribution is seldom used in applied work, perhaps due to the fact that $p(Y = 0) = p(Y = 1) = 0$. As a more popular and more plausible alternative we included the Beta-binomial distribution, which can be obtained by sampling $q$ from Beta distribution parametrized by $c_l$ and $c_u$ and $y \in 0, 1, \ldots, n$ is then sampled from Binomial distribution with proportion parameter $q$ and $n \in \mathbb{N}$. As described in [25], binomial distribution can be seen as a maximum entropy distribution with $\mathrm{E}[Y] = q$ and with additional assumptions about the discretization process.

Having presented the maximum entropy distributions, we now consider to what extent these satisfy the informal CFE conditions. As already mentioned, the second CFE condition is incorporated as an assumption in the derivation of the maximum entropy distributions, but the remaining three conditions must be checked separately. In principle, such task is in exercise of looking up the formula for expectation, variance or skew and then computing the limit.

Unfortunately, the analytic formulas for these quantities are either not known or do not exist (Generalized Gamma, Log-logistic) or the available formulas do not use the current parametrization (Beta, Beta prime, Beta-binomial, Generalized geometric). As a consequence only few results are readily available and are discussed next. The remaining results are obtained through simulation and presented in section 3.9.

In the case of Log-normal distribution, the first and the third condition are satisfied while the skew is independent of $c_l$. In the case of Beta distribution we set $c_l \to \infty$ while $c_u$ is held constant (and vice versa). As a consequence $c_l - c_u = \psi(a) - \psi((1/E[Y] - 1)a) \to \infty$. Since $\psi$ is increasing and convex, $c_l - c_u \to \infty$ when $E[Y] \to 0$ or both $a \to 0$ and $b \to 0$. In the latter case, however $c_u \to \infty$ which is in contradiction with our assumptions and hence only the former case is valid. Thus, the first condition is satisfied by Beta distribution.

Regarding Generalized Gamma distribution, note that the result depends on the choice of nuisance parameters. Trivially, if we hold $c_m = E[Y]$ constant, then $c_l \to -\infty$ will not affect the expected value. Instead, we propose to hold $c_n$ and $c_n \log b$ constant, where the latter term may be conceived as the offset of $c_l$. Then $c_l \to -\infty$ implies, $\psi(a) \to -\infty$, hence $a \to 0$ and as a consequence $E[Y] = \exp(c_n \log b)a \to 0$. The first condition is satisfied by Generalized Gamma distribution distribution.

**1.1.5 Additional distributions with CFE.**   We discuss two additional distributions that are popular in the literature, that mostly satisfy the CFE conditions and that offer a plausible connection to measurement theory, yet their maximum entropy derivation is not available in the literature.

Boundary hitting times of Brownian motion and its extensions are popular models of response times and response choice in psychology [26]. Wald distribution describes the first boundary hitting time of a one dimensional Wiener Process with one boundary located at distance $a > 0$ from the starting position of the particle, which moves with a drift $b > 0$ and a diffusion rate $s > 0$. The probability density function of the hitting time is

$$p(Y = y) = \frac{a}{s\sqrt{2\pi y^3}} \exp\left(-\frac{(by - a)^2}{2s^2 y}\right)$$

All three parameters are not identifiable. The common choice in the psychological literature is to fix $s = 1$ [26], although we prefer $a = 1$ because $s$ and $b$ are independent while $a$ on $b$ are not. The concatenation can be numerically represented by addition of positive quantities to $b$ while holding $s$ constant. The increments to $b$ increase the speed of the particle and result in a faster response.

A more recent approach by [27] at conceptualization of maximum velocity, considers the conjoint structure of velocity, duration and distance. Such conjoint structure is inherent in the process definition of Wald distribution, which suggests a possible measure-theoretic derivation of the Wald distribution. We are not aware of such derivation in the literature and we limit this section to checking whether Wald distribution satisfies the CFE conditions. The expected value, variance and skewness of Wald distribution are $1/b$, $s^2/b^3$ and $3s/\sqrt{b}$. All three quantities converge to zero for $b \to \infty$ and thus Wald distribution satisfies all four CFE conditions.

Two additional popular models for $Y \in \{0, 1, \ldots, N\}$ are the Polytomous Rasch model and Ordered logistic regression. Polytomous Rasch model is defined by

$$p(Y = y) = \frac{\exp \sum_{k=1}^{y}(c_u - c_k)}{\sum_{j=0}^{N} \exp \sum_{k=1}^{j}(c_u - c_k)}$$

where $c_u = -c_l$ represents concatenation and $c_k \in \mathbb{R}$ are threshold weights. [28] and [29] show

that Polytomous Rasch model presents an extension of Boltzmann distribution and they use Boltzmann's method of maximum probability distribution in their derivation. [19] (chapter 9) shows that this method provides identical results as POME. [18] (see section 10.5) provides POME derivation of the case $N = 3$. All these derivations utilize the constraint $c_m = E[Y]$, even though a parametrization in terms of $c_m$ is rarely used in the applied research. In any case, with $c_l \rightarrow -\infty$ and $c_k$ constant, $p(Y = 0) \rightarrow 1$ and hence $E[Y] = c_m \rightarrow 0$ and $Var[Y] \rightarrow 0$. The Rasch model thus satisfies the first three CFE conditions.

Another, different parametrization of Rasch model is available in the literature under the label *ordered logistic regression* and can be formulated as a latent variable model ([30], p. 120):

$$y = n \text{ iff } c_{n-1} > z > c_n$$

with latent variable $z \in \mathbb{R}$ generated from the logistic distribution $z \sim \text{Logistic}(c_u, c_v)$ and thresholds $c_k \in \mathbb{R}$ for $k \in \{0, 1, \ldots, N\}$. The parameters $c_u = -c_l$ provide an opportunity to manipulate the CFE. In particular $\lim_{c_l \rightarrow \infty} E[Y] = 0$ and $\lim_{c_l \rightarrow -\infty} Var[Y] = 0$. Thus Ordered logistic regression satisfies the first three CFE conditions.

The current section showed how measurement theory along with POME allows us to derive the most popular distributions with bounded or semi-bounded range. Crucially, these derivations suggest the appropriate parametrization and tell us which parameters must be manipulated to create CFE. Informal CFE conditions were considered and two additional distributions were suggested based on these conditions. This review of probability distributions will facilitate the review of robustness literature in the subsequent section and the choice and justification of the data-generating process used in the simulations as described in section 1.3.2.

## 1.2 Robustness of *t*-test and *F*-test

The previous section suggested that CFE affects mean, variance and other distributional properties of the measured quantity. These in turn, may lead to violations of assumption of normal error distribution and assumption of variance homogeneity which underlie the derivation of *t*-test and *F*-test. To what extent does violation of these assumptions affect the conclusions obtained with *t*-test and ANOVA? The early analytic derivations and initial computer simulations came to the prevailing conclusion that *t*-test and *F*-test are surprisingly robust to violations—perhaps with the exception of homogeneity violations in unbalanced designs (see [31] or a review of this early work). More recent literature however suggests that the early studies underestimated the magnitude of normality and variance homogeneity violations in data [32]. In addition, the question of robustness depends on the choice of performance measure. The decision depends on the costs and benefits of alternative methods. Early literature stressed the advantage in terms of power and computational simplicity of parametric relative to non-parametric methods. However more recent research suggests that power loss of non-parametric methods is often negligible and the advent of computer technology made use of non-parametric methods readily available.

Apart from the parametric/non-parametric distinction, large chunk of robustness studies, especially those utilizing discrete RVs, have been concerned with the danger of applying parametric methods to data with ordinal scale. As highlighted by [33], for various reasons formal measurement theory did not become popular and is rarely used in applied research. Instead a limited version of measurement theory, to which we refer as Stevens' scale types [34, 35], has become prominent. The tools of formal measurement theory provide the flexibility to design infinite variety of structures, that capture the apparently infinite variety of data generating processes encountered by the researcher. [34] proposed to reduce this complexity to four

structures (and the corresponding scales), which he considered essential. *Interval* scale preserves intervals and allows researcher to make statements about the differences between the empirical objects. *Ordinal* scale preserves ranks and allows researcher to make statements about the order of empirical objects. With ordinal scale any transformation that preserves order is permitted and does not alter the conclusions obtained from the data. However, with interval scale only linear transformations are permitted. Since *t*-test or *F*-test compute means and variances which in turn imply additive representation a controversy arose as to whether the application of parametric methods to data with ordinal scale is meaningful. Some of the related simulation studies will be described below, for a review of the issue and the arguments see [36–40]. A more recent and more specialized branch of this controversy was concerned with the application of parametric methods to data obtained with Likert scales [41–44].

Below we review the robustness studies relevant to the present topic with data generated from continuous distributions. Discrete distributions are covered in the subsequent section which is followed by our discussion of this literature.

**1.2.1 Continuous distributions.** The robustness studies of *t*-test [45–51] show a type I error change of up to 3 percentage points and a power loss of up to 20 percentage points. Most studies used the *t*-test for groups with equal variance instead of Welch's *t*-test [52] for unequal variances, which appears to be unequivocally recommended in its favour and is more robust [53–57]. For instance, [58] compared type I error rate of *t*-test, Welch test and Wilcoxon test on variates with Exponential, Log-normal, Chi squared, Gumbel and Power function distribution. Welch test showed best performance with error rate ranging to 0.083 for two Log-normal distributions with unequal variance. In this case, Wilcoxon test showed error rate of 0.491, even though it performed on par with Welch test when the variances were equal. In their meta-analytic review of the literature, though, [59] noted "the apparent sensitivity of the Welch test to distinctly non-normal score distributions" (p.334). Most simulation studies on performance of *F*-test were either performed with normal variates (see table 2 in [59] or with discrete variates (e.g. [60–63]). The few available studies with non-normal continuous distributions used exponential, Log-normal or double exponential distribution. In their meta-analysis [59] conclude that "the Type I error rate of the *F*-test was relatively insensitive to the effects of the explanatory variables SKEW and KURT" which corresponded to the skew and kurtosis of the distribution from which the data were generated in the respective studies. [56] reported similar results, but provided a results for particular distribution types (table 12 in their work). The average type I error rate across studies ranged from.048 of Log-normal to.059 of exponential distribution. At the same time KW test performance was.035 with Log-normal and 0.055 with exponential distribution. Regarding the power of *F*-test [56] did not provide an analysis as they concluded that the number of studies was not sufficient and they pointed to additional conceptual difficulties of power estimation. In particular, with non-normal distribution it's not straightforward to compute power [55]. While [59] provided an analysis of power, their analysis was inconclusive.

The studies discussed so far suggest that skew, kurtosis and choice of distribution do not notably affect the performance of *t*-test, *F*-test and KW test and other factors such as group size homogeneity and variance homogeneity of the groups are more important. These studies and reviews are not very informative in the present context as they mostly manipulate variance (type I error), or mean and variance (power) and only in few cases skew and kurtosis are manipulated. As such it is difficult make conclusions regarding the robustness with respect to CFE. Some researchers claim that the simulations are not representative with respect to the data that actually occur in research. [32] presented a survey of 440 large-sample achievement and psychometric measures. He reported that "no distributions among those investigated passed all tests of normality, and very few seem to be even reasonably close approximations to

the Gaussian." and he pointed out that "The implications of this for many commonly applied statistics are unclear because few robustness studies, either empirical or theoretical, have dealt with lumpiness or multimodality." (p. 160) [64] investigated the performance of *t*-test on data sampled from distributions that resembled those reported by [32] and concluded that *t*-test is robust in particular the "multi-modality and lumpiness and digit preference, appeared to have little impact". They further concluded that "a dominant factor bringing about nonrobustness to Type I error was extreme skew" (p.359). [64] discuss the practice of manipulating group variance and group means in isolation as part of the simulation studies intended to test type I and type II errors. They state that "we have spent many years examining large data sets but have never encountered a treatment or other naturally occurring condition that produces heterogeneous variances while leaving population means exactly equal. While the impact of some treatments may be seen primarily in measures of scale, they always (in our experience) impact location as well." We suggest that CFE is one possible mechanism that explains the dependence between mean and variance.

**1.2.2 Discrete distributions.**   Numerous studies investigated type I error rate of *t*-test and *F*-test when data were generated from discrete distribution [60–63, 65–67]. The error rate ranged from 0.02 to 0.055 ($\alpha = 0.05$) depending on the choice of data generating mechanism. These studies lack the measurement-theoretic embedding. As a consequence multiple factors such as discretness, nonlinearity, skewness or variance heterogeneity were confounded. However, some insight can be obtained from graphical and numerical summaries of the data generating process.

[60] first generated a latent discrete variable in range [1, 30] with approximately normal, uniform or exponential distribution. This was then transformed using a set of 35 nonlinear monotonic transformations. Notably, one set of transformations qualitatively resembled a floor effect (see Figure 3 in [60]) and, compared other three sets of transformations, this set manifested the worst performance, both in terms of *t*-test's type I error rate (2.8 to 5.1% with $\alpha = 0.05$) and in terms of the correlation between *t* values of transformed and raw latent values (see Table 4 in [60]).

[63] generated values from uniform, normal and normal mixture distributions which were then transformed to integers from 1 to 5 with help of predetermined sets of thresholds. [63] focused mainly on the type I error rate of Wilcoxon test and CI procedure based on *t*-test. For the present purpose of most interest is the test of difference between one group with symmetric normal distribution and another group with shifted and skewed normal distribution. *t*-test showed higher power than Wilcoxon test for small sample size. For large sample sizes both methods performed at 100% power. Again, it's not possible to judge the magnitude of power reduction due to the change of distribution properties such as skewness since a comparable control condition with shifted but not skewed condition is missing.

[67] investigated type I error rate of *t*-test with group values randomly sampled from over 140 thousand scores obtained from trauma patients with Glasgow Coma scale. The scores were integers and ranged from 3 to 15 points and their distribution manifests strong ceiling and floor effect with two modes at 3 and 15. The error rate was 0.018 for a two-tailed test with group size 10, but the error was negligible for the next larger group with 30 samples.

In addition to [63], only a couple of studies investigated the power of statistical methods with discrete distributions. We think that the reasons are conceptual. Such study requires a formulation with help of measure-theoretic concepts. As a consequence the few available studies are the ones that are conceptually most similar to the present study, even though these are not explicitly concerned with CFE. Hence these studies are reviewed in some detail.

[68] explicitly discussed formal measurement theory in relation to the controversy of application of parametric methods to ordinal scales. [68] (p. 392) argued that the question of

invariance of results under a permissible scale transformation was crucial: "A test of this question would seem to require a comparison of the statistical decision made on the underlying structure with that of the statistical decision made on the observed numerical representational structure. [. . .] If the statistical decision based on the underlying structure is not consistent with the decision made on the representational structure then level of measurement is important when applying statistical tests. Since statistical statements are probabilistic, consistency of the statistical decision is reflected by the power functions for the given test." The authors generated normal RV which was then transformed to ranks, pseudo-ranks and ranks with added continuous normal error. The authors compared the power of *t*-test between the transformed and untransformed variables. The power loss due to the transformation was rather negligible with up to two percentage points for ranks and pseudo-ranks. Power loss for ranks with noise was up to ten percentage points.

[69] generated data from uniform, normal, exponential or triangular distribution, and then squashed the values with thresholds to obtain a discrete RV with 2 to 10 levels. The authors compared the power of linear and probit regression and concluded that linear regression (OLSLR) performed well. Exponential distribution which had most resemblance to ceiling effect got a special mention (p. 383): "As an exception, we note OLSLR-based power was decreased relative to that of probit models for specific scenarios in which violation of OLSLR assumptions was most noticeable; namely, when the OCR had 7 or more levels and a frequency distribution of exponential shape."

[70] generated two group samples from ordered probit regression model and showed that a normal model may lead to incorrect inferences compared to ordered probit model fitted to these data. The ordered probit model is similar to the ordered logistic regression model described in section 1.1.5 except that $z \sim \mathcal{N}(\mu, \sigma)$. From all of the reviewed studies, the design of this study is the most similar to the current work. Note, that the authors compare the performance of parametric methods with a performance of a model which was used to generate the data. Trivially, in terms of likelihood, any model that was used to generate data will fit those data better than any other non-nested model. The current work will try to avoid such situation by using distinct and non-overlapping models for the purpose of method evaluation and for the purpose of data generation.

**1.2.3 Robustness research explicitly concerned with ceiling effect.** The research on statistical ceiling effect is difficult to find due to the abundance of research on an equally named phenomenon in managerial science. Hence the review in this section, which is focused on the research that explicitly mentions CFE in statistical sense, is likely to be incomplete.

Majority of this literature considers the Tobit model and the relative performance of linear regression and Tobit regression. Tobit model assumes that a RV $Y$ is generated as $y = y_u$ if $z \geq y_u$ and $y = z$ if $y < y_u$, where $z \sim \mathcal{N}(\mu, \sigma)$. $y_u$ is fixed and finite, while $\mu$ and $\sigma$ are model parameters. The procedure, used by Tobit, in which the value below some or above some fixed threshold is replaced with the threshold value is called *censoring*(e.g. see entry on censored observations in [10]). Censoring should be distinguished from truncation in which the values above some ceiling are discarded. The distinction between the observed $Y$ and the latent $Z$ is similar to the distinction between the empirical structure and the numerical representation, in which the concatenation is not represented by addition. In Tobit model $Z$ is an additive numerical representation while $Y$ is not. In particular, the Tobit model is an instance of a structure in which the maximal element can be obtained by concatenation and which was briefly mentioned in section 1.1.1. Contrary to the claim by [9] that such structure is not relevant to research, a considerable literature on Tobit model and CFE emerged in recent two decades.

[71] considered how ceiling effect affects the growth estimates of linear model applied to longitudinal data generated from multivariate normal model which were then subject to a threshold similar to how $Z$ was treated in the Tobit model. Up to 42% of the data were subject to ceiling effect. The authors compared performance of Tobit model with that of a linear model. The linear model was applied to data subjected to omission or list-wise deletion of the data at ceiling, which was intended to improve its performance. However, linear model performed worse and it underestimated the growth magnitude. In particular, "the magnitude of the biases was positively related to the proportion of the ceiling data in the data set." ([71] p. 491).

[13] investigated how ceiling effect changes the value-added estimates of a linear method compared to a Tobit model. Similar to [71], this was an educational research scenario with longitudinal structure, but unlike the former study, [13] used test scores obtained with human participants which were then censored. The authors concluded that Tobit model considerably improved estimation, but also concluded that the linear method performed reasonably well up to a situation in which the ceiling affected more than the half of the values. See [72–75] for additional studies on the relative performance of Tobit and linear method with similar results.

The aforementioned studies rely on censoring to generate data. The sole exception is the study by [13] who distinguish between *hard ceiling* and *soft ceiling*. Hard ceiling was generated using the Tobit procedure and corresponds to the empirical structure in which the maximal element can be obtained through concatenation of two non-maximal elements. Soft ceiling corresponds to the structure with essential maximum, which we described in detail in section 1.1.1 and which is the focus of the current work. [13] considered skewness to constitute the crucial feature of a soft ceiling and used a spline to simulate RVs with skewed distribution. The choice of spline model is ad hoc and as [13] concede there are other valid choices. [13] found that the effects of soft ceiling were somewhat less severe.

The focus on data generated with censored normal or empirical distributions, limits the conclusions of the literature discussed so far in this section. Apart from the exclusion of soft ceiling, as mentioned by [71] (p.492), there are other potential choices to model the latent variable $Z$, such as Weibull or Log-normal distribution. Another class of excluded models with similar distributions are zero-inflated models, the most notable instance being the zero-inflated poisson model [76] used in survival analysis. Furthermore, censoring is readily recognizable—especially with continuous measurements. It manifests as a distinct mode at maximum/minimum. Thus, the Tobit model provides a very specific way to model CFE, which raises the concern about generality of the results obtained with this model.

A few additional studies on CFE compare the performance of linear regression model with a generalized linear model on discrete data. [77] used logistic model to study attrition in gerontological research and claimed that the choice of generalized linear model helped them avoid incorrect inferences caused by CFE. [78] compared the data fit of four statistical models. The simulated data modelled the distribution of four psychometric tests popular in epidemiological studies. The four tested models were: linear regression, linear regression applied to square root transformed data, linear regression applied to data transformed with CDF of Beta distribution, and a threshold model similar to ordered logistic regression reviewed in section 1.1.3. The last two models provided better fit than linear regression with no data transformation. [79] compared linear regression with generalized linear model based on binomial logit link function. They analysed a large data set obtained with Health of the Nation Outcome Scale for Children and Adolescent. The scale has integer outcome in range from 0 to 52. The linear method provided considerably larger effect size estimates (change between two time point of magnitude −2.75 vs. −0.49).

The three studies of CFE on discrete data demonstrated that generalized linear model and linear regression provide different effect size estimates. Strictly, this does not imply that the

estimates from linear model are incorrect or biased. The authors made this conclusion, by assuming that the bounded values obtained with the test or the scale reflect an unbounded latent trait which seems plausible. The assumption of continuity of the latent trait and the discrepancy between the latent trait and the measured values highlights the conceptual link of these studies to the measure-theoretic framework. Unlike Tobit studies, these studies investigated the soft ceiling. All the studies reviewed in this section focused on effect size estimation of either regression coefficient or of differences (between groups or between repeated measurements) and how their estimation is affected by CFE. The effect of CFE on hypothesis testing, p values or confidence intervals was rarely discussed.

## 1.3 Goals and scope

**1.3.1 Selection of statistical methods for investigation.**   The main goal driving the selection of statistical analyses is to complement the literature by providing additional results for previously untested modern methods. We focus on performance of *t*-test and ANOVA due to their popularity. As already mentioned in the previous section, estimation of regression coefficients in linear regression under CFE has been repeatedly investigated, hence we omit the regression setting and focus mainly on hypothesis testing. In particular, we investigate the performance of Welch test as it has been recommended in favor of the standard *t*-test without any additional drawbacks. We consider one-way ANOVA with three levels and a $2 \times 2$ omnibus ANOVA, in which case we present results from two *F*-tests for the two respective main effects and the results from *F*-test for the interaction. The following alternatives to *t*-test and ANOVA are included.

First, while the robustness of non-parametric methods has been investigated extensively, the investigations did not manipulate or account for the magnitude of CFE. Instead the focus was set on the ordinality of measurement and on the distributional properties such as variance or skew. We look at the performance of Mann-Whitney (MW) Test, Kruskal-Wallis (KW) Test and Scheirer-Ray-Hare-Test (SRHT).

Second, [80, 81] argued for a wider use of so-called robust statistical methods—methods that were designed to tolerate violation of normality and presence of outliers. [82] and [83] recommended the trimmed *t*-test by [84] in which a predefined proportion of the smallest and largest data values is discarded. Investigation of the performance of the trimmed *t*-test is included in the present study.

Third, Bayesian hypothesis testing has been recommended for diverse reasons [85–88]. Notably, it allows researchers to both confirm and reject hypotheses and the results are easier to interpret than p values since they describe the probability of hypotheses. One difficulty with Bayesian methods is that they require the specification of the data distribution under the alternative hypothesis. Another difficulty is computational: nuisance parameters are handled by integrating the data likelihood over their prior probability, which, depending on the choice of prior and the amount of data, can be computationally intensive. As a consequence various approaches to Bayesian hypothesis testing have been recommended [86, 89, 90]. The Bayesian *t*-test presented by [86] was selected in the current study, due to its computational simplicity as well as to its close relation to the frequentist *t*-test, which facilitates comparison. The Bayesian *t*-test provides the probability of the null hypothesis that the groups are equal relative to the alternative hypothesis that the groups differ.

Fourth, a frequentist alternative which offers the flexibility to confirm a hypothesis is equivalence testing. In particular, "two one-sided tests" (TOST) procedure by [91] has been advocated by [92]. The procedure asks the researcher to specify an equivalence interval around the effect size value suggested by the null hypothesis. Two *t*-tests are performed to determine

whether the test statistic is respectively higher and lower than the lower and higher bound implied by the equivalence interval. Equivalence is assumed, if both hypotheses can be rejected. We are not aware of any robustness studies of the TOST procedure.

Fifth, a shift from null-hypothesis significance testing to reporting of (standardized) effect sizes along with confidence intervals was advocated by numerous authors (see e.g. [93–95]). Since any hypothesis testing procedure may be used to construct a confidence interval, the decisions based on confidence intervals fail under conditions similar to $t$-test. [96] showed with data generated from skewed distributions that the capture rate of the parametric CI does not correspond to the nominal CI. As an alternative, they recommended a CI of a bias-corrected effect size computed with bootstrap procedure. In the current work we present effect size estimates along with CI estimates in a research scenario with two groups. We present both the unstandardised effect size which corresponds to the mean difference in the observed value and its standardized version—Cohen's $d$. The effect size estimates and CIs can be viewed as an inferential alternative to $t$-test, but in the present context they also highlight how CFE affects mean, variance and standard error, i.e. the elementary quantities utilized by the $t$-test computation.

Sixth, the literature review suggested that extreme skew and heterogeneous variances are (in addition to unequal group size) crucial factors leading to poor performance of $t$-test. The review of measurement theory as well as some publications [13] suggest that the two are related and we investigate the issue further by providing estimates of mean, variance and skew of the generated data. This further facilitates the discussion of the results as it helps us to relate these to those obtained by robustness studies with focus on skew manipulation.

Finally, our goal was to evaluate a statistical inferential method that would specifically target CFE and provide an alternative to the established methods. In selecting such method we had two concerns. First, the method should be reasonably easy to apply in terms of both computation and interpretation. Second, we wanted to avoid the obvious but not very helpful choice to use models identical to the data-generating process. This would mean for instance, that we compare the performance of generalized gamma model with say a $t$-test when the data are generated from generalized gamma distribution. Such setup has been used previously (e.g. [70]), and it may be useful for demonstrating drawbacks of the linear method, where by fitting a model identical with the data-generating mechanism one obtains an upper bound on the performance. However, as an alternative to established methods, it's not helpful to recommend a model that precisely matches the data-generating distribution, as the main difficulty is that the data-generating mechanism is not known, or only very rough knowledge is available. Such rough knowledge may be a suspicion that CFE affects the measure. The candidate statistical methods can be justified by considering the measurement theory behind CFE that was discussed in section 1.1.1. There we considered the appropriate models of CFE and for each distribution we distinguished the nuisance parameters from parameters describing the magnitude of CFE. The latter correspond to $c_u$ and $c_l$ in Table 1. We saw that in most cases CFE is described by a logarithmic function which is further modified by the nuisance parameters e.g. $c_l = \log\left(Y^{c_n}/c_m\right)$ in the case of Gamma distribution. Thus, one may assume that the logarithmic function provides accurate descriptions of CFE even though its precise shape determined by the nuisance parameters may not be known. We think that this situation, better describes the applied case, in which the researcher may suspect an influence of ceiling effect due to the bounded measurement range or due to the skew; but she may not have the exact knowledge of how CFE affects the measurement.

Given these consideration we decided to use Log-normal model and Logit-normal model in scenarios with one and two bounds respectively. In fact, since these models correspond to an application of normal linear model to data transformed by log or logit function, we make the comparison more direct by comparing t-test with t-test applied to log or logit transformed

data. The data transformation is additionally used in the case of one-way ANOVA, two-way ANOVA, with CIs, Bayesian $t$-test and equivalence testing, thereby obtaining an additional version of each statistical method. In our opinion, such design helps us distinguish to what extent the performance of each statistical method depends on the assumption of normality.

To conclude, the selected statistical methods used in the current work are Welch's $t$-test, one-way ANOVA, two-way ANOVA, Bayesian $t$-test and TOST. Each method is applied to log-transformed or logit-transformed data. In addition, $t$-test is compared with trimmed $t$-test and rank-based MW test, while one-way ANOVA is compared to KW test and two-way ANOVA is compared to SRHT.

**1.3.2 Selection of data-generating process and performance metric.** The present work will utilize Beta distribution, Generalized Gamma distribution, Beta prime distribution, Wald distribution, Beta-Binomial distribution and Ordered Logit Regression model (OLRM). We excluded the Log-normal and Logit-normal distribution as these would coincide with the models which performance is being investigated. The discrete Beta distribution and generalized geometric distribution listed in Table 1 are excluded for lack of applied interest in these models. In addition, theoretical results needed for simulation of such random variables are not available in the statistical literature.

Even more important than the choice of distributions is the choice of their parametrization. All distributions are parametrized with $c_i$ as described in section 1.1.3. Gamma distribution is the sole exception, for which we use parameter $b$ instead of $c_m$. The latter choice would result in the expected value $c_m$ being held constant, which in turn would result in a trivial failure of any statistical method which uses the difference between means for inference. We gave priority to OLRM over polytomous Rasch model as the former appears to be more popular being included in textbooks such as [30] and has generated some prior research interest relevant to the current topic [70, 78, 79].

Since all three distributions which manifest both ceiling and floor effect use the same function for ceiling and floor effect ($f_u = f_l$), without a loss of generality, we restrict the investigation to a floor effect and the related parameter $c_l$. Then the goal is to investigate the ability of statistical methods to detect constant difference in $c_l$ as the offset of $c_l$ goes to $-\infty$. In particular, we investigate seven research scenarios. First, two groups (A and B) of 50 values each are generated from the respective distributions with $c_l^A = c_l^0$ and $c_l^B = c_l^0 + c_l^\Delta$. We manipulate the magnitude of CFE by varying $c_l^0$, while $c_l^\Delta$ (which may be interpreted as an effect size) and all remaining parameters are held constant. The two-group scenario is used to investigate the inference with Confidence intervals, with Cohen's $d$, Welch's $t$-test, Trimmed $t$-test, MW test, TOST, Bayesian $t$-test and Welch's $t$-test on transformed data. In the second situation, we introduce a third group with $c_l^C = c_l^0 + 2c_l^\Delta$ and the performance of one-way ANOVA, KW test and one-way ANOVA on transformed data is investigated. The remaining five scenarios assume presence of two factors with two levels each and and the performance of $2 \times 2$ ANOVA applied to raw and transformed data and the performance of the rank-based SRHT is investigated. Fig 2 illustrates the five different types of interactions that are being investigated. The choice of the scenarios is motivated by the research on how CFE affects linear regression, which showed that the linear method has problems identifying interactions [78]. In the first two cases we are interested in whether the statistical methods are able to correctly distinguish between a main effect and an interaction. In the first case, one main effect occurs and there is no interaction. The second case describes a situation with interaction but without main effect. The last three cases include two main effects and an interaction on $c_l$, but differ in the order of the $c_l$ mean values of the four groups. We label these uncrossed, crossed and double-crossed interaction. Assuming that CFE affects the performance, the double-crossed interaction should pose least problems for

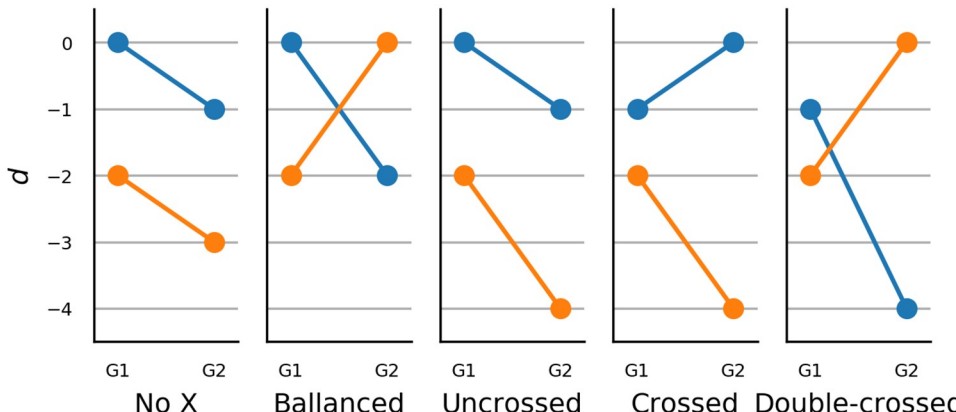

**Fig 2. Types of interactions.** Each panel shows a qualitatively different outcome of a 2 × 2 factorial design. The first panel (from left) shows two main effects without an interaction. The second panel shows an (additive) interaction but no main effect. The remaining three panels show different constellations of two main effects accompanied by an interaction. Crucially the relative order of the conditions differs across these three constellations. The five displayed outcomes are used in the simulations.

statistical methods, since CFE preserves order information and the inferential methods should be able to pick out this information. In contrast, the uncrossed interaction should pose most problems. As in previous scenarios, the group differences are held constant, while $c_l^0 \to -\infty$. In each interaction case the group specific $c_l$ is computed as $c_l^C = c_l^0 + c_l^\Delta d$ with $d$ shown in Fig 2. The procedure is repeated for various combinations of nuisance parameters. Notably, for each distribution we look at how different values of $c_l^\Delta$ affect performance. One additional nuisance parameter is investigated for each distribution. The identity of these nuisance parameters as well as the ranges of all parameters are listed in Table 2.

Regarding the robustness metric, we repeat each simulation 10000 times and present the proportion of cases in which a group difference was detected. Assuming a binomial model, the width of the 95% CI of the proportion metric is less than 0.02. As an exception, when reporting the performance of confidence intervals, Cohen's d and Bayesian *t*-test, we report median values. With Bayesian *t*-test we report the median probability of hypothesis that the two groups differ as opposed to the hypothesis that the two groups are equal.

## 2 Materials and methods

### 2.1 Design

For each of the eight distributions and each of the seven scenarios, two, three or four groups of data were generated with 50, 33 or 25 samples per group respectively. The statistical methods

**Table 2. Parameter values used in simulation.**

| Distribution | Range $c_l$ | $c_l^\Delta$ value | $c_l^\Delta$ range | NP label | NP value | NP range |
|---|---|---|---|---|---|---|
| Gen. Gamma | [−4, 4] | 0.25 | [0, 1] | $c_n$ | 1.0 | [0, 2] |
| Wald | [20, 0] | 1.0 | [0, 4] | $\sigma$ | 1.0 | [0, 2] |
| Beta Prime | [−10, −2] | 2.0 | [0, 8] | $c_m$ | 0.155 | [0, 0] |
| Beta | [−20, 0] | 3 | [1, 5] | $c_u$ | -0.2 | [0, 0] |
| Beta-Binomial | [−5, −1] | 4 | [1, 8] | $n$ | 7 | [3, 15] |
| OLRM | [−5, 5] | 0.8 | [0, 1] | $n$ | 7 | [3, 15] |

NP- nuisance parameter

were then applied and the inferential outcome was determined and recorded. This was repeated 10000 times and the performance metrics were computed over repetitions. Furthermore, the procedure was repeated for 25 combinations of five values of $c_l^A$ and five values of a distribution-specific nuisance parameter. The distribution-specific nuisance parameter is listed in Table 2 (column NP label). The five values of $c_l^A$ and of the nuisance parameter were selected from the ranges listed in Table 2 with equally spaced intervals.

## 2.2 Software implementation

The implementation relied on the stats library from Python's Scipy package (Python 3.7.3, Numpy 1.16.3, Scipy 1.2.1). A translation from the $c_l$ and/or $c_u$ to the more common parametrization of Gamma and Beta distribution (in terms of $a$ and $b$ in Table 1) was obtained iteratively with Newton method. The formulas for compution of the gradient are provided in [97] (chapter 7.1) and [98] (chapter 25.4) respectively. To compute the initial guess we used the approximation $\psi(x) = \log(x - 0.5)$.

The confidence intervals were computed with the normal-based procedure (theorem 6.16 in [99]). The standard error was that of the Welch test. To compute standard error of Cohen's $d$ we used the approximation in [100] (Eq 12.14): $\mathrm{SE}_d = \sqrt{\frac{2}{n} + \frac{d^2}{4n}}$ where $n$ is the sample size of each group. The denominator in computation of Cohen's $d$ was (Eq 12.12 [100]): $\sqrt{\frac{(n-1)(v_1 + v_0)}{2n-2}} = \sqrt{\frac{v_1 + v_0}{2}}$ where $v_i$ is the group variance. The TOST thresholds were fixed across $c_l$ to correspond to an median estimate of the group difference obtained with a $c_l$ located 3/4 between the lower and upper bound of the $c_l$ range. E.g. $c_l$ used to obtain the threshold for gamma distribution was $4 - (-4) \times 0.75 - 4 = 2$. Separate thresholds were obtained for transformed and raw data. The threshold estimates were obtained with Monte Carlo simulation with 5000 samples in each group. The code used to run the simulations and to generate the figures is available from https://github.com/simkovic/CFE.

## 2.3 Parameter recovery with ordered logistic regression and Beta-binomial distribution

This section describes the methods used in section 3.8. The motivation behind this supplementary investigation is described in section 3.8. In the first step OLRM was fitted to ratings of moral dilemmas reported in [1] and [2]. The data are labelled $y_{n_s i}^{gs}$, where $y \in 0, 1, \ldots, 9$ is the rating, $n_s$ indexes the participant, $i \in \{1, \ldots, 6\}$ indexes the item, $g \in \{0, 1\}$ indicates whether the data come from the experiment group or from the control group, and $s \in \{0, 1\}$ indicates wheter the data come from the original or from the replication study. OLRM is then given by $y_{n_s i}^{0s} \sim \mathrm{orlm}(b_i^s - c_k)$ for control group and for experimental group it's $y_{n_s i}^{1s} \sim \mathrm{orlm}(b_i^s - c_k + d_s)$ where $c_k \in \mathcal{R}$ is the set of nine thresholds, $b \in \mathcal{R}$ is item difficulty and $d_s$ is the group difference.

In the second step a set of data points $z_{nc_u g}$ were generated with $z_{nc_u g} \sim \mathrm{orlm}(c_u + gd_0 - c_k)$, where $d_0$ and $c_k$ were fixed to median estimates of the corresponding parameters obtained in the previous step. $c_u$ was fixed to one of 50 values in the range $[-7.7, 4.8]$ with equally spaced intervals between the values. In the third step another OLRM was fitted $z_{nc_u g} \sim \mathrm{orlm}(b_{c_u} + gd_{c_u} - c_{kc_u})$ with which estimates of $b_{c_u}$, $d_{c_u}$ and $c_{kc_u}$ were obtained.

The estimation was performed with PySTAN 2.19 [101] which is a software for statistical inference with MCMC sampling. In each analysis, six chains were sampled and a total of 2400 samples was obtained. The convergence was checked by estimating the potential scale

reduction $\hat{R}$ ([102] p.297)in the parameters. In all analyses and for all parameters $\hat{R} < 1.05$ where $\hat{R} = 1$ upon convergence.

## 3 Results

The results are presented as graphs with $c_l^0$ on the horizontal axis and performance metric on the vertical axis. To facilitate comparison, statistical methods which were applied to the same data, in the same scenario and with the same metric are presented together in the same graph. Excluding $2 \times 2$ ANOVA, the result is a total of 1050 graphs, which we structure as 42 figures with $5 \times 5$ panels. It is neither feasible nor instructive to present all figures in this report. The figures are included in S1 Appendix. Most of the graphs manifest qualitatively similar patterns and in the results section we present, what we deem to be a fair and representative selection. In particular we present for each distribution the results for a particular pair of values of $c_l^\Delta$ and of the nuisance parameter. These values are listed in the third and the sixth column of Table 2.

### 3.1 Generalized gamma distribution

Fig 3 displays the results for Gamma distribution with $c_n = 1$ and $c_l^\Delta = 0.25$. In the first panel, the raw group difference converges to zero and the CI gets narrower as $c_l$ decreases. Accordingly, as $c_l$ decreases, Bayesian $t$-test and TOST switch to support the group equivalence. In contrast, the difference between log-transformed group data oscillates around the true value, with CI getting wider as $c_l$ CL decreases. Accordingly, log-TOST does not suggest equivalence for low $c_l$, however the Bayesian log-$t$-test still switches to incorrect conclusion and shows only marginal improvement compared to Bayesian $t$-test on raw data. Cohen's $d$ goes towards zero with decreasing $c_l$ for both raw and log data. Accordingly, $t$-test and log $t$-test both fail to detect a group difference for low $c_l$, with log-$t$-test showing better performance of up to 6.9 percentage points (pp) over raw $t$-test. The rank-based test manifests similar performance (up to 3.2 pp difference) as log-$t$-test, while trimmed $t$-test shows biggest gap to log-$t$-test with up to 20 pp. The test performance in the three group design provides similar results, with $F$-test trailing by up to 10.1 pp behind log-$F$-test and the rank-based test, while all three tests show a switch from group difference detection to no detection as the floor effect increases in magnitude.

Other choices of $c_l^\Delta$ and $c_n$ do not affect the qualitative nature of the results. Note that $c_n$ and $c_l^\Delta$ function both as scale with respect to $E[logX]$. When statistical methods are applied to log-transformed data, then, a different choice of $c_l^\Delta$ and $c_n$ merely scales $c_l$ and stretches or shrinks the horizontal axis in Fig 3. With respect to $E[X]$, higher values of $c_l^\Delta$ and $c_n$ make CI tighter, increase the mean group difference but also affect the shape of CI. The performance gaps between the tests increase as $c_l^\Delta$ and $c_n$ increase with the absolute test performance increasing as well.

### 3.2 Wald distribution

Fig 4 displays the results for Wald distribution with $\sigma = 1$ and $c_l^\Delta = 0.25$. Note that the severity of floor effect increases as $b$ increases. We inverted the horizontal axis in Fig 4 to facilitate comparison between figures. CI of the group difference decreases towards zero with both raw data and with the log-transformed data. Both CIs become narrower as the magnitude of floor effect increases. The CIs of Cohen's $d$ are identical for both types of data, with Cohen's $d$ decreasing towards zero. As a consequence the $t$-test and log $t$-test (and MW test as well) show similar performance. Trimmed $t$-test performs worst with a gap of up to 20% to $t$-test. All two group and three group tests manifest a decrease in performance as the magnitude of the floor effect

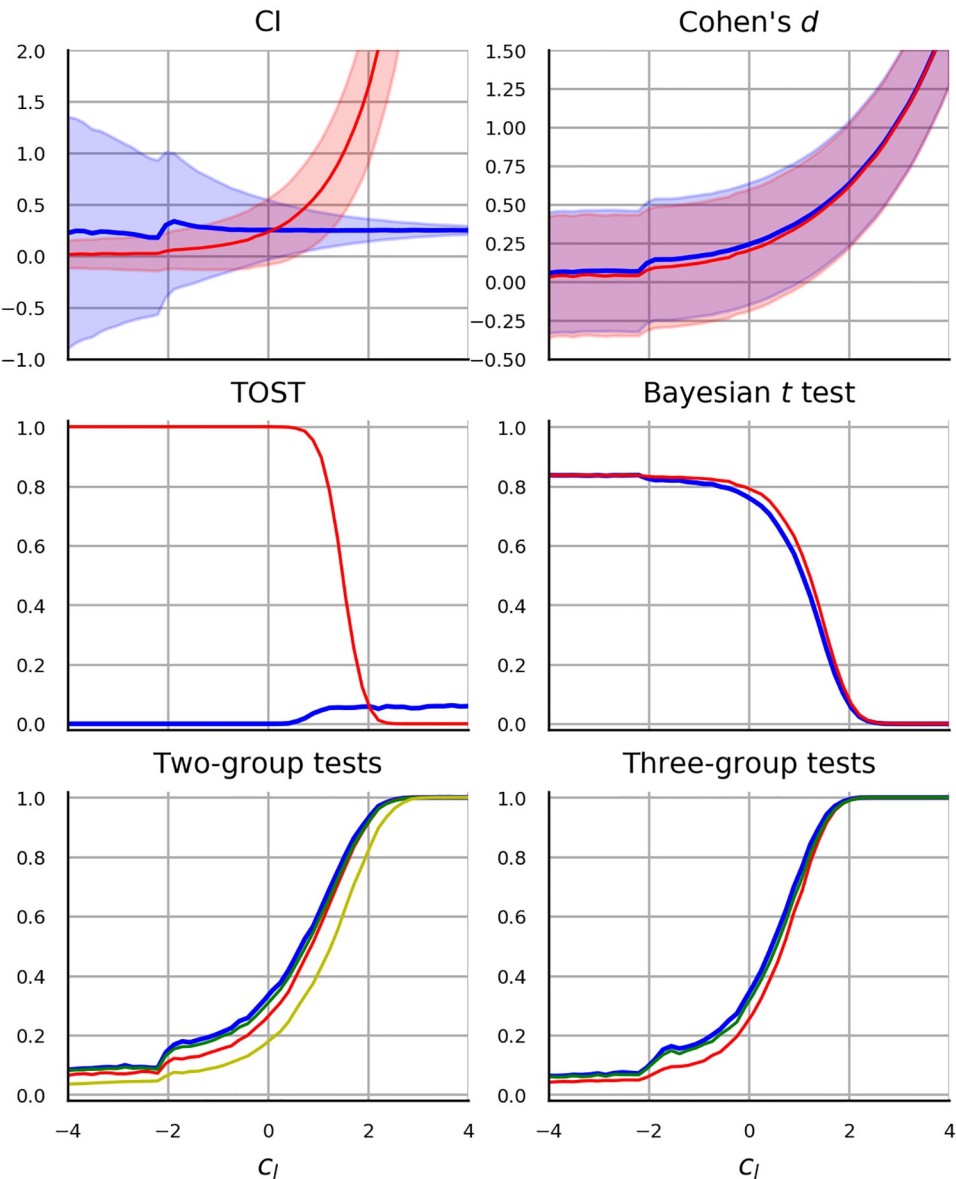

**Fig 3. Results obtained with Generalized Gamma distribution.** The panels are referred to in order from left to right, from top to bottom. Horizontal axis in each panel shows $c_i$ ($c_i^A = c_i^0$ to be precise). The colored polygons in the first panel show the mean group difference (thick line) and its 95% CI (surface). The blue color shows results with log-transformed data while the red color shows results with raw data. The second panel shows Cohen's $d$ along with CIs. The layout is similar to the layout of the first panel. The third panel shows results of the TOST equivalence testing procedure. The vertical axis shows the proportion of cases which supported group equivalence. In the fourth panel, the vertical axis shows the probability of the hypothesis of group mean equivalence relative to the hypothesis of a group mean difference. The blue and the red color in the third and fourth panel designate whether the methods are applied to transformed or raw data. The vertical axis in the fifth and sixth panel shows the proportion of rejections of null hypothesis (group equivalence). Fifth panel shows results from $t$-test (red), log-$t$-test (blue), MW test (green) and trimmed $t$-test (yellow). The sixth panel show results from three-level-$F$-test with transformed data (blue) and raw data (red). The green line shows results of KW test. The values in panels one, two and four were obtained as median across replications.

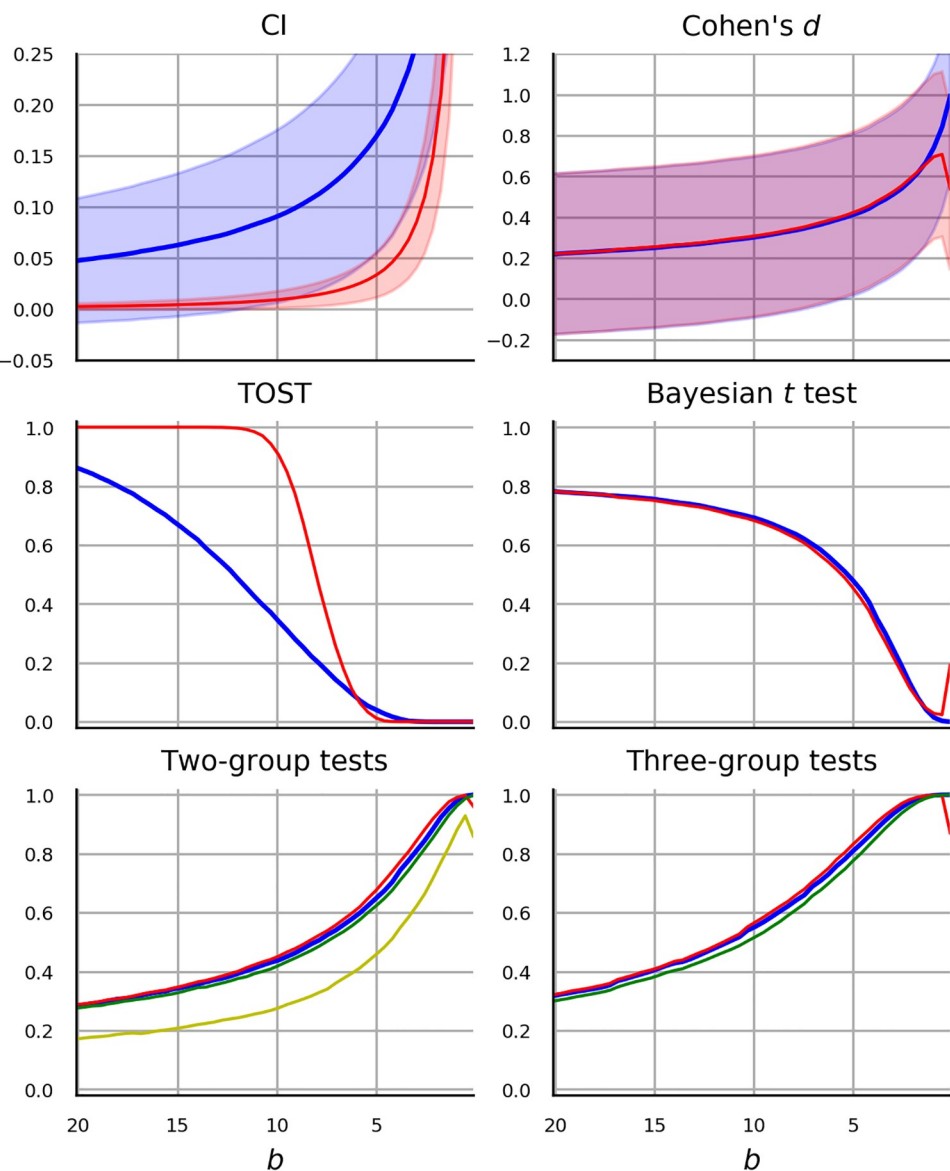

**Fig 4. Results obtained with Wald distribution.** The figure layout follows the layout of Fig 3. Refer to Fig 3 for details.

increases. As the floor effect becomes stronger, both TOST and Bayesian $t$-test switch to support group equivalence.

Adjustments to $\sigma$ and $c_i^\Delta$ alter the scale and offset of $b$ but otherwise do not change qualitative nature of the results. Higher values of $c_i^\Delta$ and lower values of $\sigma$ lead to better test performance and gaps between the tests are larger when both values are large. These adjustments also suggest the test performance does not converge to zero. The performance at the convergence point depends on $\sigma$ and $c_i^\Delta$.

### 3.3 Beta prime distribution

Fig 5 shows the results for Beta Prime distribution with $c_n = 0.15$ and $c_i^\Delta = 2$. The mean group difference is negative but close to zero, while the mean group difference of log-scaled

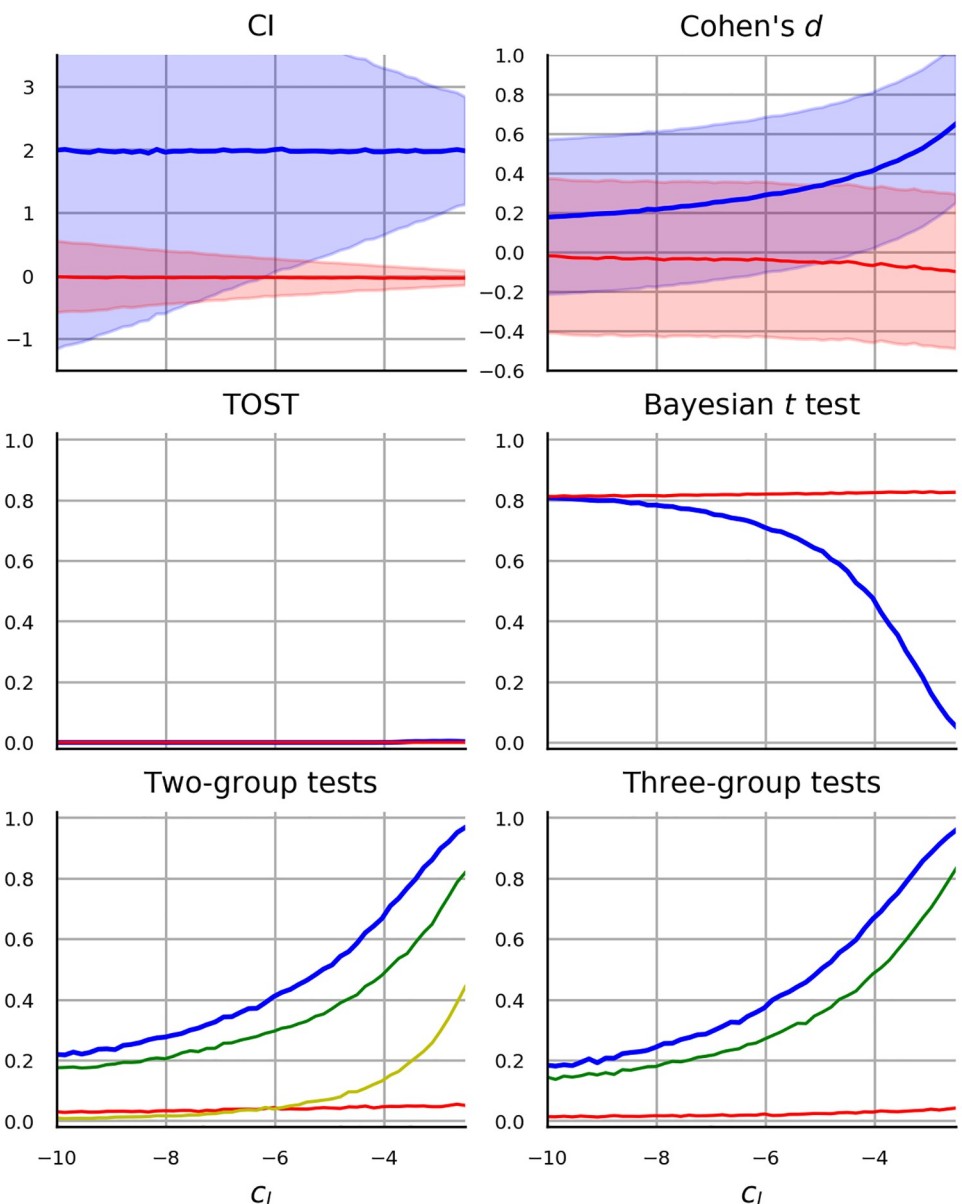

**Fig 5. Results obtained with Beta Prime distribution.** The figure layout follows the layout of Fig 3. Refer to Fig 3 for details.

data oscillates around the true value of $c_i^A$. In both cases the CI becomes wider as $c_l$ decreases. The test performance in the two-group and in the three group scenario decreases with decreases in $c_l$. The performance of $t$-test and $F$-test with raw data is in the range $[0, 0.1]$ irrespective of $c_l$. Similar, Bayesian $t$-test indicates support for no difference from zero with raw data irrespective of $c_l$, but with log-data Bayesian $t$-test switches from support of difference to support of no difference. Log-$t$-test and log-$F$-test show best performance, which is followed by rank-based tests (gap of up to 20.1 and 19.4 pp respectively) and by trimmed $t$-test (gap of up to 60.8 pp). TOST rarely detects equivalence because the TOST threshold is too conservative.

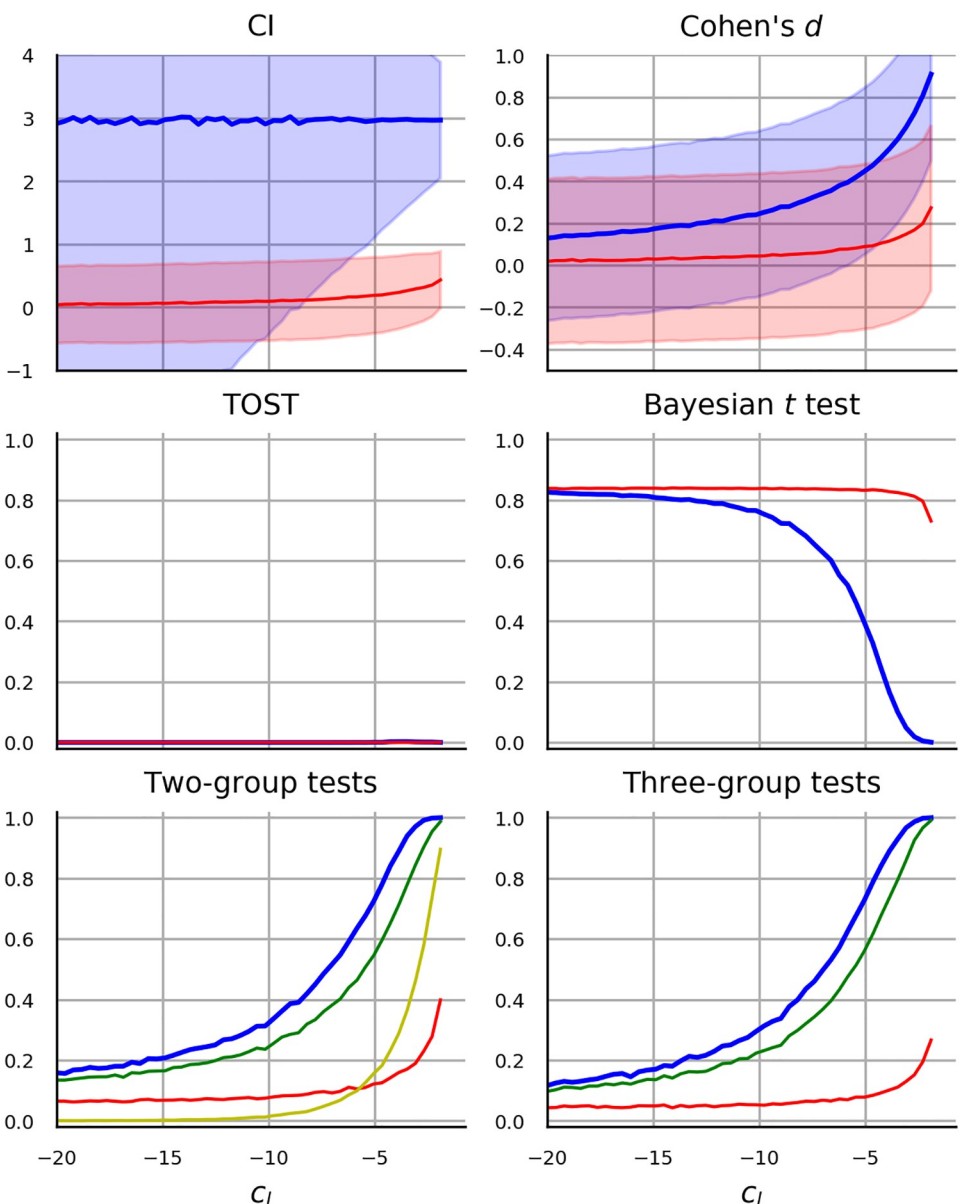

**Fig 6. Results obtained with Beta distribution.** The figure layout follows the layout of Fig 3. Refer to Fig 3 for details.

Large values of $c_n$ and $c_l^A$ increase the (negative) mean difference and the width of the CI. The CI of the log-transformed data is unaffected. Finally, larger $c_l^A$ result in better test performance, including the performance of Bayesian $t$-test on log-scaled data.

### 3.4 Beta distribution

Fig 6 shows the results for Beta distribution with $c_u = -0.2$ and $c_l^A = 3$. Recall, that in this case logit transform is used instead of log transform. The difference in mean of logit values oscillates around the true value while with raw data the difference decreases towards zero with decreasing $c_l$. In the former case the CI width increases as the magnitude of floor effect increases, while in the latter case it remains constant. Cohen's $d$ of both raw and logit data

decreases with decreasing $c_l$ while the corresponding CI remains constant. The logit $t$-test shows best test performance followed by rank based test with a gap of up to 19.1 pp and raw $t$-test and trimmed $t$-test with even larger gaps. In all four cases, a switch from high rejection rate to low rejection rate is observed as $c_l$ decreases. The scenario with three groups shows similar results. Bayesian $t$-test switches to support equivalence rather than group difference as $c_l$ decreases. As in previous section, the TOST thresholds are too conservative to detect equivalence. However a switch to support equivalence can be observed when TOST is applied to data generated with larger values of $c_l^\Delta$.

Larger $c_l^\Delta$ and smaller $c_u$ improve the test performance. Smaller values of $c_u$ make CIs narrower while $c_l^\Delta$ increases the mean difference. In all cases, the logit-based difference estimate matches the true group difference $c_l^\Delta$.

### 3.5 Beta-binomial distribution

Fig 7 shows the results for Beta distribution with $c_u = -0.3$, $n = 7$ and $c_l^\Delta = 4$. In this case the blue color highlights a transformation with logit function. For both transformed and raw data, the CIs are similar: the mean decreases with decreasing $c_l$ while the CI width remains constant. Similar pattern is observed with Cohen's $d$. The rank-based test performs best with a gap of up to 25.3 pp to logit $t$-test. The logit $t$-test shows better performance than the raw $t$-test when the data are unaffected by floor effect. Trimmed $t$-test performs worst with a gap of up to 49.9 pp to rank-based test. Similar results are obtained in the three group scenario, with rank-based test showing the best performance. Larger $c_l^\Delta$ and larger values of $n$ improve the test performance. Both $n$ and $c_l^\Delta$ increase the mean difference, while higher $n$ results in wider CI.

With more severe floor effect, Bayesian $t$-test favours hypothesis of no group difference. TOST once again does not detect equivalence as the thresholds are too conservative.

### 3.6 Ordinal logistic regression model

Fig 8 shows the results for OLRM with $\sigma = 0.8$ and thresholds at $-3, -1, 1, 3$. Again, blue lines show results for data transformed with logit function. In this case $c_l = -c_u$. We used this fact to show both floor and ceiling effect in Fig 8. As $c_l$ decreases, floor effect increases, while as $c_l$ increases, the ceiling effect increases in magnitude. Note that one of the groups was further offset with respect to $c_l$ on the horizontal axis, which explains why the graphs are not fully symmetric around $c_l = 0$. With ceiling and floor effect the positive mean difference goes to zero and the CI becomes narrower. This is true for both raw and transformed data as well as their Cohen's $d$. The test performance decreases as the magnitude of ceiling and floor effect increases. Regarding test comparison, all tests with exception of trimmed $t$-test show comparable performance. TOST supports equivalence at floor and at ceiling. Similar, Bayesian $t$-test favours equivalence at ceiling and at floor. Note that in principle the simulation could be extended beyond $c_l = -5$ and $c_l = 5$, however at $c_l = -5$ already more than 90% of values are zero, and the presence of the floor effect should be apparent to an investigator.

### 3.7 Two-way factor design

Fig 9 shows the comparison of detection rate of rank-based SRHT (green) with ANOVA (red) and with ANOVA on transformed data (blue). Log transform is used with positive outcomes in the first three columns. Logit transform is used with bounded outcome in fourth, fifth and sixth column. Dotted lines show the stronger main effect, dashed line shows the weaker main effect and full line shows the interaction. The vertical axis shows the proportion of rejections of hypothesis that there is no main effect/interaction. As in previous section, horizontal axis

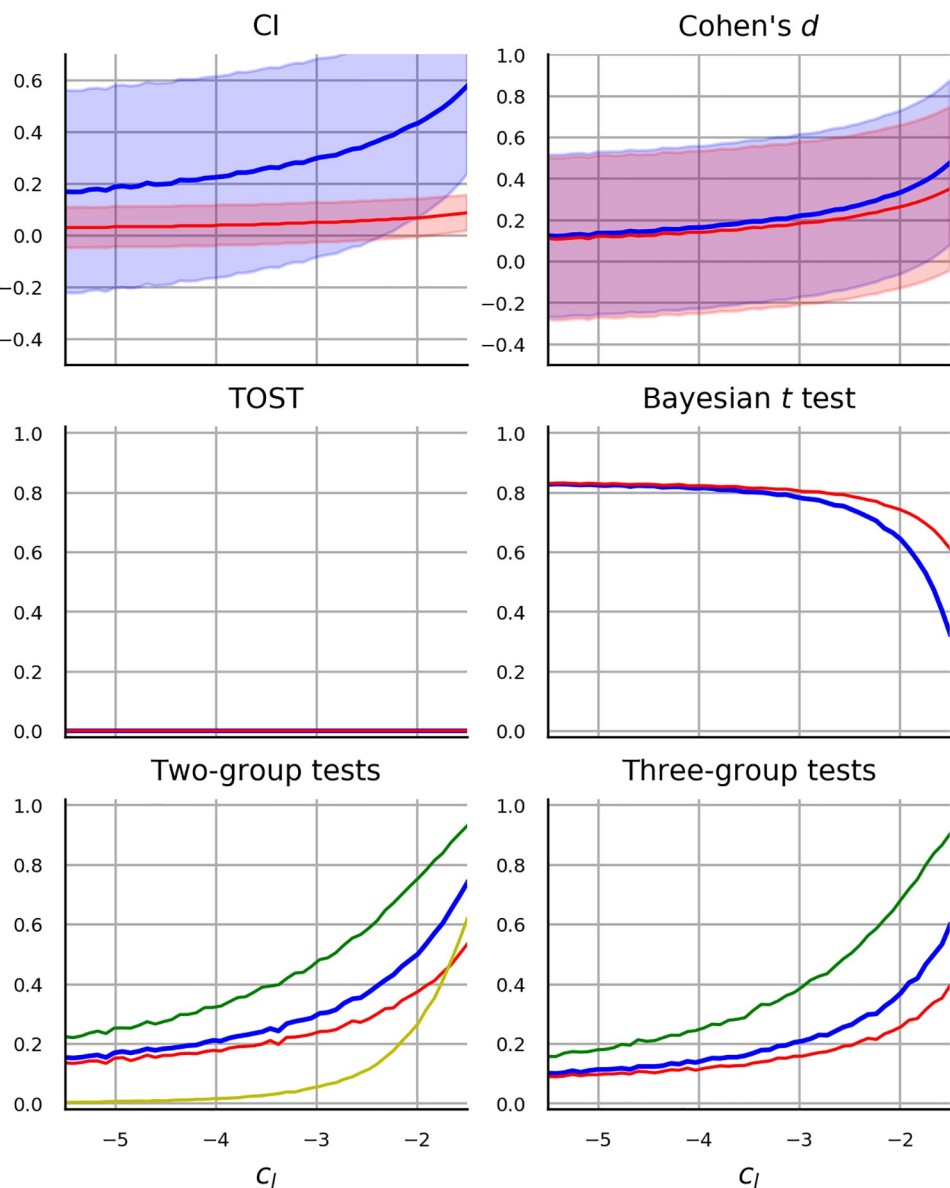

**Fig 7. Results obtained with Beta-binomial distribution.** The figure layout follows the layout of Fig 3. Refer to Fig 3 for details.

shows the values of parameters used to create floor effect. The floor effect increases in the negative (to the left) direction of the horizontal axis. This is true of Wald distribution as well, in which case we reversed the values of $b$ such that higher values are located towards the left part of the axis. The panel columns show six different data generating processes, while the rows show five different main effect/interaction constellations. These constellations were described in Fig 2. Most importantly, recall that in the "no X" situation there were two main effects but no interaction, "no ME" labels situation with interaction but no main effects and in the remaining three situations both main effects and an interaction occurred.

First, note that over all conditions the detection performance decreased as the magnitude of the floor effect increased. In the "no X" situation there were two main effects but no

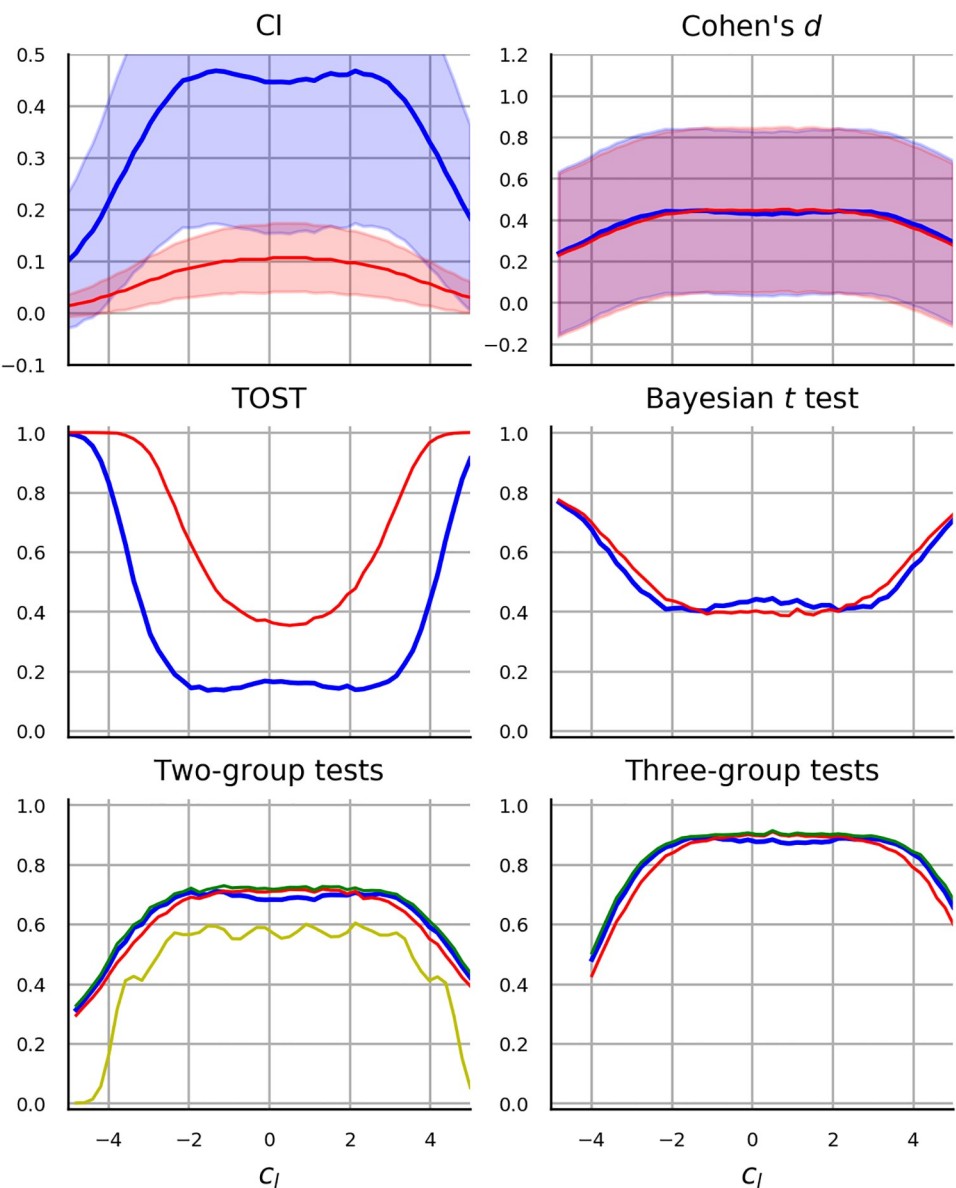

**Fig 8. Results obtained with OLRM.** The figure layout follows the layout of Fig 3. Refer to Fig 3 for details.

interaction. The tests mostly managed to detect the main effects when the floor effect was weak. The tests correctly avoided detection of interaction in most cases. As an exception, *F*-test (with raw data) detected interaction when the floor effect was weak while the data were generated from Gamma or Wald distribution. *F*-test furthermore failed to detect the stronger main effect with Beta prime and Beta distribution (columns 3 and 4), which was in contrast with the other two methods. SRHT outperformed the other two tests on Beta-binomial data, while with Beta prime and Beta distributed data the log *F*-test showed slight advantage over SRHT.

"no ME" labels the situation with an interaction but no main effects. All methods correctly avoided detecting main effects. Raw ANOVA encountered problems with detecting interaction when the data were generated from Beta and Beta prime distribution. In these cases the

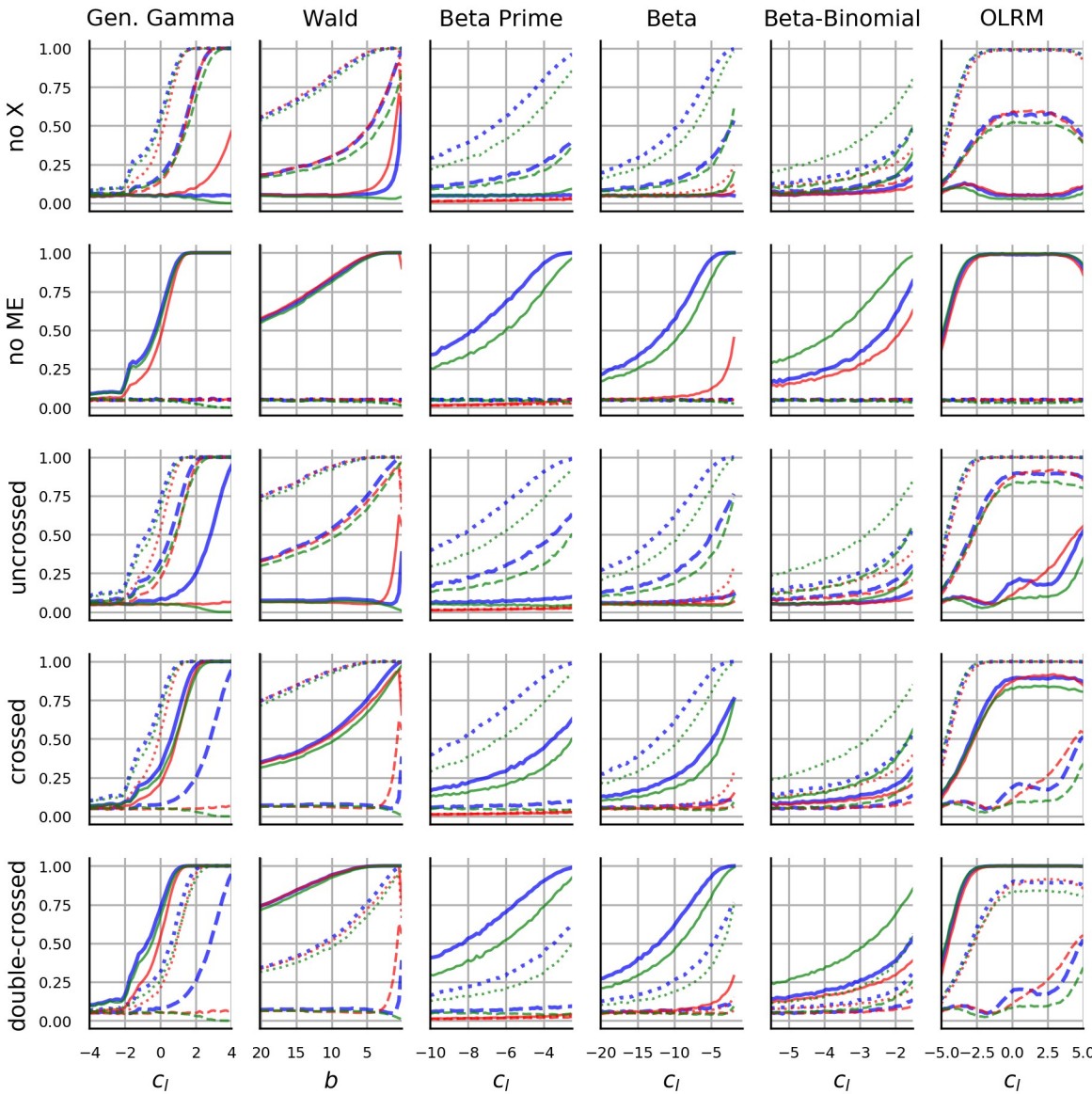

**Fig 9. Performance with 2 × 2 factor design.** Figure shows the comparison of detection rate of rank-based SRHT (green) with ANOVA (red) and with ANOVA on transformed data (blue). Further details are provided in the text.

log $F$-test displayed a power advantage over SRHT. In contrast, SRHT was most successful at detecting interaction when data were generated from Beta-binomial distribution.

In the remaining three situations both main effects and an interaction occurred. Note that the three results in the last three rows are very similar, except they show different permutations between the two main effects and the interaction. Compare the third and fourth row: if one exchanges the dashed and the full lines, one obtains results that are very similar. Between the fourth and the fifth row, the dotted and full lines are swapped. This is perhaps not surprising as the three different interaction types were obtained by swapping the true mean values between factors.

All three statistical methods encountered problems detecting uncrossed interaction. The sole exception was log-ANOVA on Gamma distributed data when the floor effect was weak.

Accordingly, all tests (again with exception of log-ANOVA on Gamma distributed data) encountered problems detecting the weaker main effect (dashed lines) in the presence of crossed and double-crossed interaction. Regarding main effects that accompanied the uncrossed interaction, ANOVA with transformed data showed best performance followed by rank-based test and ANOVA. Again, when the data are generated from Beta Prime and Beta distribution $F$ test with transformed data showed advantage over SRHT, while SRHT benefited in the case of Beta-binomial distribution. In all these three cases the raw $F$-test showed poor performance at detecting the main effects. When data were generated from Ordered Logistic model, the three tests showed similar performance.

In the scenario with no interaction ("no X") and with uncrossed interaction when the data were generated from Beta, Beta prime and Beta-binomial distribution, the reader may wonder what the interaction detection performance would look like, if one would select larger $c_l$. Due to the constraints of the beta function such choice is not possible, one may however increase the effect size i.e. scale the group differences to facilitate easier detection. The last figure in S1 Appendix shows the detection rate in the $2 \times 2$ factor scenario with $c_l^\Delta$ set to the upper bound listed in the fourth column in Table 2. The figure indicates that for large $c_l$ in "no X" scenario SRHT provides at least some support for interaction with data from Beta, Beta prime and Beta-binomial distribution. The interaction detection performance of the other two methods remains unaffected. The larger $c_l^\Delta$ does not improve detection of uncrossed interaction for these three distributions. Interestingly, raw $F$-test outperforms log $F$-test when $c_l^\Delta$ and $c_l$ are large.

## 3.8 Parameter recovery with Ordered logistic regression and Beta-binomial distribution

As shown in Sections 3.5 and 3.6, the application of logit transformation to discrete data did not provide a performance improvement relative to methods applied to untransformed data. Such result raises the question, whether this outcome describes an ineffectiveness of the selected transformation or whether the discrete data present a more general inferential challenge, that requires a rank-based solution. In this section we provide clarification by fitting OLRM and Beta-binomial distribution to data generated from OLRM. As already mentioned in the introduction, the difficulty with this procedure is computational one. The analytic results for parameter estimation with OLRM or Beta-binomial distribution under the parametrization used in the current work are not available in the literature. Approximate methods are available but computationally expensive. For these reasons, parameter recovery (i.e. the same model is used to generate and fit the data) was avoided in the previous sections even though it would provide interesting information about the best-case performance. For these reasons, the current section does not consider performance across 10000 repetitions and nor does it consider performance across a range of nuisance parameters. Only a single research scenario is considered. To make the choice of the scenario less arbitrary and less artificial, we adapt the research scenario discussed in the introduction. We ask how the magnitude of ceiling effect would affect the conclusions regarding the replication attempt by [2]. In particular, we fitted OLRM to the data from [1] and [2]. We estimated the difficulty separately for each of the six items in the original study and we estimated a separate set of parameters for the replication study. The item difficulty of the control group was $c_u$ while the item difficulty of the experimental group was $c_u + c_u^\Delta$ The thresholds $c_k \in \mathbb{R}$ were pooled across the items and across the datasets. The group difference $c_u^\Delta$ was identical across items but two separate parameters were used for original and replication sample. In the second step, multiple sets of fake data were generated from the OLRM with thresholds and with the group difference fixed to the median

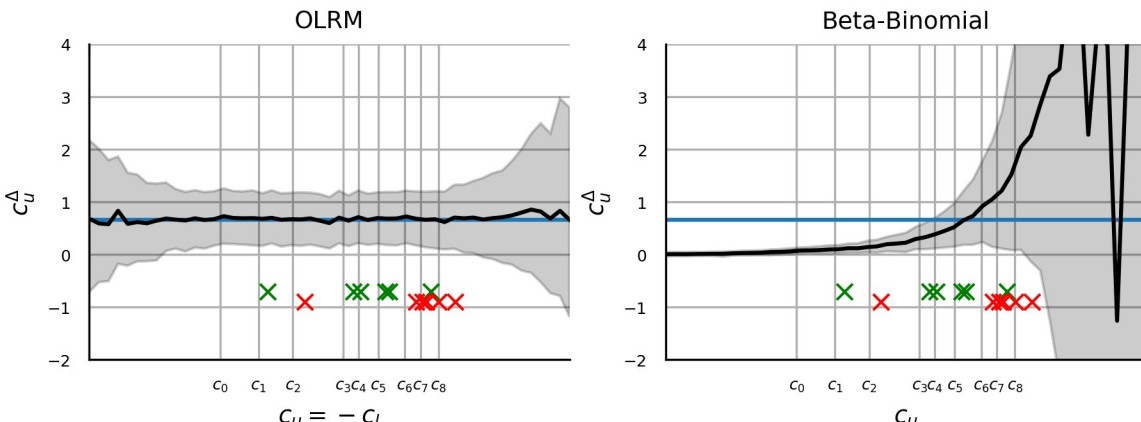

**Fig 10. Parameter recovery with Ordered logistic regression and Beta-binomial distribution.** Figure shows the estimate of group difference $c_u^\Delta$ on vertical axis that was obtained by fitting OLRM (left panel) and Beta-binomial distribution (right panel) to the data generated with OLRM by manipulating the magnitude of ceiling effect $c_u$. $c_u$ is shown on the horizontal axis and the magnitude of the ceiling effect increases from left to right. In reference to the precise notation in section 2.3, the black line shows the median estimate of the group difference $d_{c_u}$, while the gray surface shows the estimate's 95% interval. The blue line shows the true value used by OLRM that generated the data and the true value corresponds to an OLRM estimate of group difference in [1] (i.e. median estimate of $d_0$). The crosses show the difficulty of the six items (i.e. $b_i^s$) in the original study (green [1]) and in the replication study (red [2]), while $c_i$ and grid lines of the horizontal axis show the OLRM thresholds obtained by pooling the original and replication data.

estimates obtained from the original study in the previous step. Similar to the procedure that was used to obtain the results in section 3.6, $c_u$ was varied in order to adjust the magnitude of the ceiling effect. OLRM was fitted to each fake data set. Note that a separate $c_u$, $c_u^\Delta$ and a separate set of thresholds was estimated for each fake dataset. Since no repetitions were available, only the median estimate of the group difference $c_u^\Delta$ along with 95% percentage interval is shown. Markov chain Monte Carlo method was used to obtain the estimates. The technical details of this method are provided in section 2.3.

The left panel of Fig 10 shows the estimate of the group difference (vertical axis) as a function of the ceiling effect which increases from left to right. The median estimate (black) matches the true group difference (blue). The width of 95% interval (gray surface) increases as the magnitude of the ceiling effect (and of the floor effect as well) increases.

As already argued in the introduction, parameter recovery can be useful for estimating the best-case performance, but we don't think it demonstrates superiority of the fitting/generating model over some other statistical model fitted to the same data. Thus the results in the left panel of Fig 10 can't be used to argue for superiority of OLRM's performance over the performance of, say, linear methods that were considered in section 3.6. Hence we added a final investigation in which the group difference was estimated with Beta-binomial model with parameters $c_l$ and $c_u$. The results are shown in the right panel of Fig 10. The median group difference is not constant and does not match the true group difference of OLRM. However, similar to the OLRM estimate, the width of the percentage interval increases as the magnitude of the ceiling effect increases.

The crosses in Fig 10 show the difficulty of the six items in the original study (green [1]) and in the replication study (red [2]), while the ticks on the horizontal axis show the OLRM thresholds obtained by pooling the original and the replication data. Items were more difficult in the replication study than in the original study. To return to the question from the opening of the introduction: does the stronger ceiling effect in the replication study mask the significant group difference? If we consider the question whether the probability that the group difference $c_u^\Delta$ is smaller than zero, is smaller than 0.025, then the answer is no. Across all items the lower

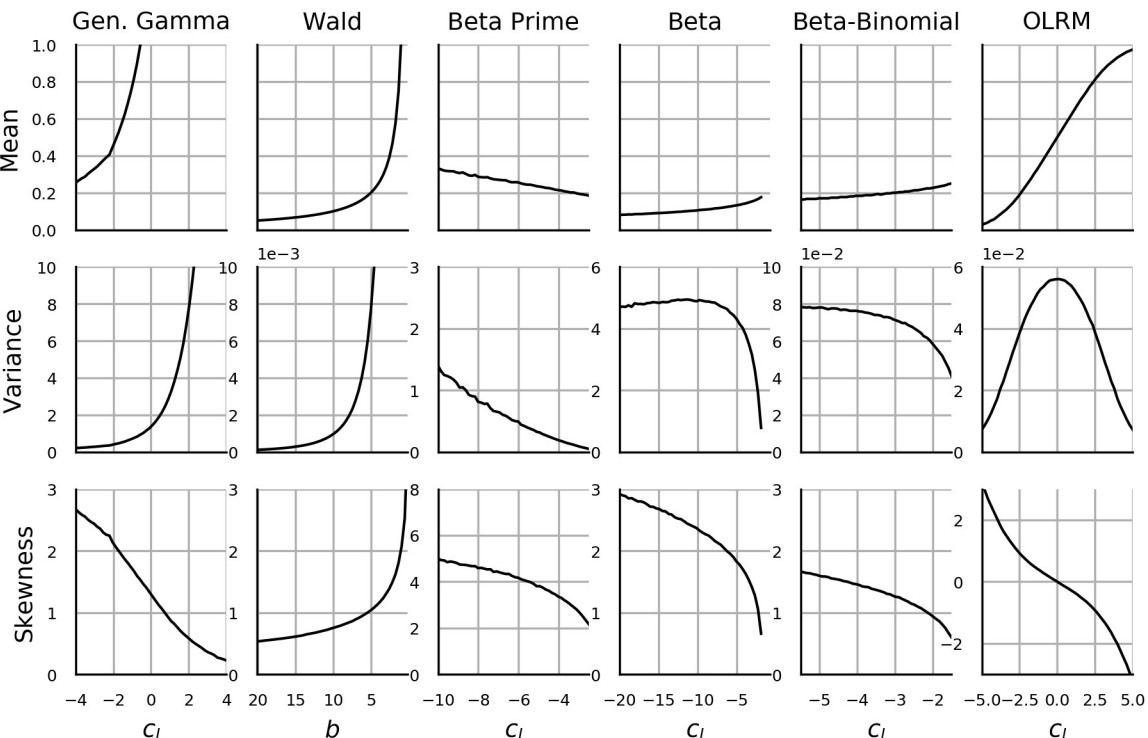

**Fig 11. Effect of CFE on mean, variance and skewness.** Figure shows how the magnitude of floor effect affects the expectation, variance and skewness of the distributions shown in columns. In each panel, the magnitude of the floor effect increases from right to the left. Further details are provided in the text.

bound of the OLRM percentage interval does not cross zero for all levels of $c_l$ that correspond to the item's difficulty. This is true for $c_u^{\Delta}$ of Beta-binomial model, with the exception of the most difficult item. If the original and the replication study consisted of six items with difficulty of the sixth item, then the significant group difference would disappear in the replication study due to the ceiling effect. However, considering the actual difficulty of the items in the replication study, in the context of estimation with Beta-binomial distribution, such disappearance is highly unlikely.

## 3.9 Mean, variance and skewness

Fig 11 shows how the magnitude of floor effect affects the expectation, variance and skewness of the distributions shown in columns. Note that OLRM manifests ceiling effect as well as floor effect, because $c_l = -c_u$. To facilitate the comparison, the values obtained with Beta distribution, Beta-binomial and OLRM were scaled to interval [0, 1]. With exception of Beta prime distribution the mean decreases as $c_l$ decreases. The variance goes to zero for Gamma distribution and Wald distribution while in the case of Beta prime distribution the variance increases with decreasing $c_l$. To consider Beta and Beta-binomial distribution, as $c_l$ decreases, the variance initially increases, reaches maximum at $c_l = c_u$ (which corresponds to $a = b$) and then decreases as $c_l < c_u$. The variance of OLRM shows maximum at $c_l = 0$ while the skew is zero at this point. As $c_l \rightarrow -\infty$ the variance goes to zero and the distribution manifests a positive skew. As $c_l \rightarrow \infty$, the distribution shows a negative skew and its variance goes to zero. With the exception of OLRM, the skew is positive. For all distributions except the Wald distribution, the skew is increasing as $c_l$ decreases.

The distributions mostly satisfy all CFE conditions. One exception is Wald distribution which manifests decreasing skew and thus fails the fourth condition. Beta Prime distribution provides the complementary case, it satisfies only the fourth condition and fails the remaining conditions. Note that the second condition can be checked by considering the mean difference shown in first panel of the preceding figures from results section. All mean differences, with exception of Beta Prime distribution, are decreasing with decreasing $c_l$.

## 4 Discussion

### 4.1 Bias and uncertainty due to CFE

We first organize and digest the presented findings. When considering the performance of inferential methods we find it advantageous to distinguish the categories of inferential outcomes listed below. These categories are not meant to be exhaustive or precise. The categories are intended to help us refer to certain phenomena and to facilitate the subsequent discussion.

**Correct inference.** One may imagine an ideal case in which there is no statistical bias (i.e. the magnitude of CFE does not alter the mean group difference) and no added uncertainty due to CFE (i.e. the magnitude of CFE does not affect the width of CI). Such scenario was not encountered in the current study.

**Noisy inference.** In this scenario, CFE increases uncertainty but does not cause bias. Noisy inference is perhaps an intuitively acceptable outcome: if the measurement tool fails to provide accurate information, it is desirable to recognize the situation as such and to avoid over-confident inferences. All of the considered inferential methods do offer an inferential outcome option which implies a noisy inference scenario. Exceptions are *t*-test, *F*-test and rank-based tests, where a failure to reject null hypothesis, may signal both inferential uncertainty *and* a situation in which the null hypothesis is true. Noisy inference is apparent from confidence intervals that were obtained with log and logit transformed data from Gamma, Beta and Beta Prime distribution. Noisy inference was further found in the investigation of parameter recovery with OLRM which was reported in the left panel of Fig 10. This results also suggests that noisy inference is the best-case scenario, when the uncertainty about the functional formulation of the parametric model is removed.

**Biased noisy inference.** In this scenario CFE creates both bias and increases uncertainty. In most cases CFE caused a bias of the mean difference towards zero. Biased noisy inference with bias towards zero was observed with untransformed data from Beta and Beta prime distribution. Furthermore, biased noisy inference with bias towards zero was observed with transformed data from Beta-binomial distribution. When fitting the Beta-binomial model to data from OLRM, CFE amplified the group difference.

Biased noisy inference may be considered as a fortunate outcome, since the increasing uncertainty prevents a biased conclusion. Similar fortunate outcome occurs in scenarios with bias and large but constant uncertainty and hence we include these in the category of biased noisy inference. Estimates of Cohen's *d* and its CI and perhaps the mean difference estimates with raw data from Beta and Beta-binomial distribution manifest bias towards zero which is dominated by noise.

**Biased inference.** Biased inference is the worst inferential outcome, in which the inferential method supports the incorrect conclusion and the inferential method becomes more certain about the incorrect conclusion as CFE increases. All biased confidence intervals that become narrower as magnitude of CFE increases fall into this category. Such is the case with raw CIs of Gamma distribution, Wald distribution and OLRM as well as with log CIs of Wald distribution and logit CIs of OLRM.

**Test performance gap.** As already stated, the performance of *F*-test and *t*-test do not fit well in the above mentioned categories, because they do not permit a distinction between bias and uncertainty, since a failure to reject null hypothesis may be caused by each of these or their combination. Rather the outcomes differ in what we label as test performance gap. In most test cases the detection rate goes from one to zero as the CFE magnitude increases. The tests differ in terms of the onset of this degradation. As the performance starts to degrade, gaps between the tests open up. All test scenarios manifest such a gap perhaps with the exception of Wald distribution and OLRM, where the performance gaps are too small to consider them as reliable. In most cases when such power gap opens up the test with transformed data performs best, followed closely by rank-based test and followed by a wider margin by test on raw data and the trimmed *t*-test. A notable exception is the case of Beta-binomial distribution in which the rank-based test performs best followed by a transformation-based test. Note that this applies across all three research scenarios: two groups, three groups and $2 \times 2$ factorial design.

It should be noted that similar power gaps were observed in power studies that investigate the robustness of parametric and non-parametric methods, see for instance figures in [46, 58, 103]. In these studies, the standardized effect size was drawn on the horizontal axis, which suggests that standardized effect size and the magnitude of CFE are related. The top-right panel in Figs 3, 4, 5, 6, 7 and 8 confirms that this is the case. In all cases where a power gap opens up, Cohen's d monotonically increases with $c_l$.

**Poor test performance irrespective of CFE.** In the case of Beta Prime distribution the mean is inverse proportional to $c_l$ and hence the *t*-test and *F*-test (in both three group and four group situation) do not open up a gap but rather consistently fail to reject the null hypothesis irrespective of the magnitude of CFE.

## 4.2 Performance of statistical methods when affected by CFE

The transformation-based and measurement-theory-motivated methods outperformed other methods with data from Gamma, Beta and Beta prime distribution and matched the performance of other methods with data from OLRM and Wald distribution. This can be observed in terms of performance gap in the two-group, in the three-group scenario and in the $2 \times 2$ factor design. It can be also observed in terms of bias and uncertainty pattern manifested by CIs. Relative to the raw data application, transformation-based methods showed more often the more benevolent biased and biased noisy inference rather than biased inference.

With data from Beta-binomial distribution, the rank-based tests showed best performance. It is worth noting that in the case of Gamma, Beta and Beta prime distribution the choice of transformation function was derived from measurement theory while the measurement-theoretic considerations of Wald distribution suggested that CFE does not follow a log function. The choice of logit transform on discrete data did not take into account the discreteness. We think this is the main reason why logit transform failed to replicate the success of log and logit transform when these were applied to continuous data. The results in section 3.8 suggest that it is possible to obtain noisy inference (OLRM) or biased noisy inference (Beta-binomial) on discrete data with with the help of a generalized linear models which implicitly assumes existence of a non-additive concatenation operation. Whether these models can match the rank-based tests in terms of test performance can't be determined from the current data and requires further research.

One may as well ask whether the rank-based tests are able to match the performance of transform-based *F*-test when applied to continuous data. Is there an information loss associated with rank-transformation and CFE? The rank-based tests manifest test performance gap to the transformation-based tests on data from Beta and Beta Prime distribution which

suggests that there is a loss of information and performance. A similar performance loss of rank-based tests was observed in comparisons with *t*-test when the data were generated from normal distribution [46, 55].

Welch's *t*-test with raw data showed mixed results. For OLMR and Wald distribution it matched the performance of other tests, while for Gamma distribution the performance gap to log *t*-test was small. For Beta-binomial and Beta distribution the gap was wider and for Beta Prime distribution Welch test failed to reject null hypothesis irrespective of CFE due to the mean inversion. Beta Prime distribution did only satisfy one of the four CFE conditions, so perhaps it may be discarded as an odd result. With raw data the *F*-test in three-group design and the *F*-tests in the two factor design show a performance similar to *t*-test.

One should note at this point that the relative and absolute detection performance of MW test and raw *t*-test are similar to the power estimates in previous robustness research [51, 64, 103]. As mentioned in the introduction, several robustness studies generate data with exponential distribution while the (standardized) group difference was varied. Recognizing that exponential distribution is a special case of generalized gamma distribution and that, in the current study, the effect size varied proportionally with the magnitude of CFE, it is valid to make a comparison. One still needs to take into account that most earlier robustness studies used *t*-test that assumed equal group variance, while the current study uses Welch's *t*-test. We are not aware of comparable robustness studies that focus on Wald, Beta prime, Beta, Beta-binomial distribution or some special case of these. The few available studies with OLRM do investigate the performance of linear regression models rather than hypothesis testing performance.

On few occasions, poor test performance was encountered with large values of $c_l$. In "no X" scenario with Gamma distribution and large values of $c_l$, the raw *F*-test provided some support for interaction and this support was stronger with larger $c_l^A$. In "no X" scenario with Wald distribution and large values of $c_l$ raw *F*-test and log *F*-test provided some support for interaction. SRHT failed to detect uncrossed interaction of Gamma and Wald distribution (and accordingly failed to detect the weaker main effect in the scenario with crossed and double-crossed interaction). Finally, raw *F*-test failed to detect an interaction with Beta prime distribution irrespective of the underlying interaction type and raw *t*-test failed to detect group difference in the two-group and three-group scenario. Since these problems affect large values of $c_l$, one needs to ask whether these problems are caused by CFE or by some other factor. Notably, one may argue that these failures are caused by the nonlinear numeric representation of concatenation. There certainly are nonlinearities that do not manifest as CFE and one would expect, that these nonlinearities affect the performance irrespective of the magnitude of CFE. However, if we accept the definition of CFE as essential maximum/minimum or if we use the formal CFE conditions as defining conditions, such definitions do *not* imply that the amount of nonlinearity should covary with the magnitude of the CFE or in fact that for large $c_l$ the nonlinear behaviour should disappear. This is in contrast with the Tobit maximum/minimum definition of CFE, where concatenation translates into addition for values that are far away from the boundaries. Gamma distribution is a special case. We provided its derivation in terms of essential minimum, but, as argued by [24], Gamma distribution can be viewed as a distribution with two regimes where for small mean values it behaves like power-law distribution (which can be derived with the POME constraint $c_l = E[\log(Y)]$) and for large mean values it behaves like normal distribution. In the latter case one would expect concatenation to map to addition while in the former case one would expect concatenation to map to multiplication. Since for fixed $b$ the expected value of Gamma distribution covaries with $c_l$ one would expect that the amount of nonlinearity increases with the magnitude of CFE. This may as well explain why

raw $F$-test was more successful at detecting uncrossed interaction when $c_l^A$ was large, while log $F$-test performed better when $c_l^A$ was small. The log transformation functions well with power-law distribution, but with data from normal distribution one would prefer an analysis of untransformed data in order to detect an interaction. We return to the issue of relation between CFE and nonlinearity in the subsequent section. For the current purpose, we conclude that the relation is not simple one and that the occurrence of poor performance with large $c_l$ depends on how well the measurement-theoretic assumptions behind the statistical method match the data generating mechanism.

Trimmed $t$-test showed consistently the worst performance of all four two-group tests. Trimmed $t$-test showed large performance gap to other tests and only occasionally outperformed $t$-test with untransformed data. Apparently, distribution tails include important information and discarding this information when the data are affected by CFE results in a poor performance.

Performance of TOST can be largely deduced from the CI pattern. A bias towards zero in the difference estimate is accompanied by TOST support for equivalence irrespective of whether it is a case of biased or a case of biased noisy inference. In contrast, with noisy inference, TOST does not support equivalence. For half of the distributions (Beta, Beta prime, Beta-binomial) the TOST threshold was apparently too large so that TOST did not support equivalence irrespective of CFE magnitude. As described in the methods section, the TOST thresholds were fixed to correspond to an median estimate of the group difference obtained with a $c_l$ located 3/4 between the lower and the upper bound of the $c_l$ range. Separate thresholds were obtained for transformed and raw data. Simple rules for obtaining thresholds are not available in the literature and we chose this threshold selection procedure to select thresholds that are plausible and so that we can compare the performance when TOST is applied to transformed data with when it is applied to untransformed data. Perhaps, it would be possible to make a more informative selection on a case-by-case basis. This would raise concerns about the plausibility of such choice and about the comparability of results. As such, a bigger investment into the derivation and the justification of such thresholds would be required. To maintain the focus we omitted a detailed investigation and leave it to future research.

As the magnitude of CFE increased, Bayesian $t$-test invariably favoured the equivalence, irrespective of the distribution and whether or not a data transformation was used. This was even the case when CIs showed noisy inference. In an ideal case, one would expect the probability of the Bayesian $t$-test to converge to 0.5 and thereby to recognize the uncertain situation as such. This did not occur. It's worth noting that Bayesian $t$-test by [86] assumes equal group variance which explains its poor performance.

Across all conditions, Cohen's $d$ decreased as the floor effect increased. Such result is unproblematic, if it is understood that Cohen's $d$ depends, in addition to the theoretical construct which one intends to measure, on the adequacy of the measurement tool and possibly on the base level at which the difference is measured. Unfortunately, the recommendations regarding the use of effect size for meta-analysis and experiment planning (e.g. [95]) take this rarely into account.

## 4.3 Ordinality, discreteness, nonlinearity and skewness

As argued in the introduction, we consider CFE to be one of the crucial phenomena that account for the performance loss reported in the robustness literature. At the same time the discussion in the published work is focused on other factors such as ordinality, nonlinearity, discreteness or skewness The current results permit further consideration of the relative importance of the these factors.

The performance of logit transform on discrete data fell below that of log and logit transform with continuous data, both in terms of failing to achieve noisy performance and in terms of performance gap to rank-based tests. At the same time we demonstrated that noisy inference can be obtained with discrete data when OLRM was fitted to data from OLRM. Fitting Beta-binomial model to data from OLRM showed biased noisy performance, but the bias was to overestimate rather than to underestimate the group difference. If the research goal is to test whether the mean group outcomes are equal, then such bias is not problematic. Apart from pointing to the inadequacy of logit transform's application to discrete data, these results suggest that when the measure offers an information from the non-additive representation of concatenation, then a statistical method designed to take into account this information, outperform the rank-based and linear methods (see [70, 78] or similar results). Whether such information is available from some particular measurement tool is an empirical questions that requires further investigation. While studies that compare the fit of linear and nonlinear models to data suspected of CFE exist (e.g. [72, 78]) and show the better fit of nonlinear methods, we are not aware of model fitting comparison between nonlinear and rank-based methods on data with CFE. In sum, these results suggest that discreteness is an important aspect of data, that should not be ignored when designing and selecting statistical tools with help of measurement theory. A recourse to rank-based methods may alleviate the problems due to discreteness, but will discount the information offered by concatenation operation. If such information is available in the measure, its omission can lead to lower power and to a confusion regarding the presence or absence of an interaction.

Another factor, that is often considered during comparisons of rank-based and linear methods as well as in studies with Tobit model, is the nonlinearity of the data-generating process, or to put it more precisely, the measure's non-additive representation of concatenation. Due to the non-additive representation, *F*-test confused interactions and main effects in the four-group scenario and this was already discussed in the previous section. There we discussed the case of Gamma distribution which switches between linear and nonlinear regime. Thus, the view of nonlinearity as some fixed aspect of the data-generating process may be misleading. While there may be other causes of nonlinearity, based on the data surveys in [32] it appears that, when it comes to psychological data with bounded range, CFE is the prevalent cause of nonlinearity. [75] noted that linear regression underestimates the regression coefficients of interaction terms when tested on data from Tobit model. It should be noted that the nonlinear behaviour of Tobit model is slightly different. When the data are far away from Tobit threshold, the data show linear behaviour. As the data approach threshold, the nonlinearity manifests and causes a misdetection of main effects and interaction. Finally as CFE gets more severe, any effects become indiscernible. The CFE created by essential minimum/maximum does not manifest the last stage.

Finally, skewness has been discussed by [13] as the main aspect of CFE and multiple robustness studies focused on skewed data. The current work further showed that the selected distributions with CFE show association between floor effect and a positive skew. Wald distribution was an exception, in which the skew decreased with the increasing severity of the floor effect thereby suggesting that the relation between skewness and CFE is not as straightforward. Perhaps, a probabilistic formulation of a structure with essential minimum implies a positive relation between skewness and the magnitude of floor effect, while the velocity-based account behind Wald distribution implies a negative relation. Further research into the connection between these measure-theoretic structures and the associated probability distributions is required to answer this question. Apart from skew, the results suggested that variance decreases with CFE. Beta prime distribution proved to be an exception with increasing variance and the variance of Beta and Beta-binomial distribution decreased only slowly.

In sum, CFE describes a constellation of several phenomena, such as heterogeneous variance, strong skew or nonlinear relation between measurement and the latent trait. The measure discreteness may add to that. The overview of the robustness literature suggested that these factors are detrimental to the performance of popular inferential methods. The current study illustrated, that when these phenomena co-occur, the resulting performance loss is not just sum of its parts, but ranges from cases of biased noisy inference, in which the detrimental effects cancel out, to cases of biased inference in which the detrimental effects reinforce each other. Hence, these phenomena need to be considered in conjunction.

A constellation of phenomena, that commonly finds reference in the literature, is the concept of ordinal scale. The measure-theoretic motivation behind this concept has been surveyed in the introduction. Unfortunately, due to the popularity of Steven's scale typology, ordinal scale is not used to refer to a measurement tool that exclusively utilize the order information. Rather all measures with an ordered value range, that do not fit into the category of nominal scale are lumped together as ordinal. We think this does not only pose a problem for theoretical discussions, but it actually affects the presentation and interpretation of simulation studies. For instance, the study by [70] is a prima facie investigation of CFE with methods and results similar to the current study. The authors even highlight the cases where the popular statistical methods fail and these cases manifest the strongest CFE. However, the authors fail to notice this, they never refer to ceiling or floor effect in their publication, and rather frame their study as a robustness investigation with ordinal data.

To name other instances, [65] and [47] created ordinal data by manipulating variance, skew and kurtosis of a continuous variable, which was then transformed into a discrete variable with hundred levels. The discretization with so many levels had negligible effect on the test performance, and the values were mostly allocated in the middle of the scale. As such these studies investigate, under the label of ordinal data, the effect of variance, skew and kurtosis on the robustness. In contrast, [63] generated "ordinal data" with five discrete levels, and while skew and variance was manipulated, most probability mass was allocated to the inner three levels so that CFE was not a concern. Thus, this was a study that investigated discreteness under the label of ordinality. Finally, [68] compared test performance of methods applied to continuous data with performance of methods applied to ranks obtained from the continuous data. Thus, this study generates ordinal data in the measure-theoretic sense. As one may imagine, the conclusions of these studies diverge, which is perhaps the reason why the application of *t*-test and *F*-test to ordinal data remains a controversial topic [40–43, 70, 104, 105] even after decades of accumulated research. We believe that the robustness research would benefit from switching its focus from the blurry concept of ordinal scale, to other more specific concepts such as skewness, nonlinearity, discreteness, or as we already suggested, to focus on their co-occurrence as ceiling and floor effect.

### 4.4 Limitations, scope and future research

**Focus on qualitative results.** In the results section, we did not dwell much on quantifying the results. We pointed the reader to figures and occasionally mentioned the maximum performance gap between a pair of statistical methods. Such qualitative approach is not uncommon to power studies (e.g. boneau62), although for instance [51] extensively report the maximum performance gaps in tables. The main purpose of the latter type of reporting is to compare conditions in terms of power advantage/loss or to design recommendations and guidelines such that the applied researcher may estimate the power loss based on simple statistical descriptors derived from the data set at hand. Clearly, the current study is not suited to provide such recommendations as that would require the knowledge of $c_l$ and of values of other distribution

parameters. For the same reason we find comparisons between conditions difficult. Can we, for instance, conclude that the performance loss of $t$-test relative to $t$-test with transformed values is more severe for Beta than for Gamma distribution, because the performance gap that occurred in Fig 6 was larger than that in Fig 3? This depends on the choice of $c_l^\Delta$ (for each Gamma and Beta distribution), $c_u$ (Beta), $c_n$ (Gamma) and possibly on the choice of additional parameters not considered in this work (such as $b$ of Gamma). As we noted the values of these parameters affect the size of the performance gap and in principle could be used to find a scenario with gap of up to 100 pp. In contrast the qualitative patterns (such as biased, noisy and biased noisy inference; or misidentification of interaction) occurred irrespective of the choice of nuisance parameter and of $c_l^\Delta$. The few deviations that occurred were discussed in the results section.

**Type I error.** We avoided to frame the current simulations in terms of Type I and Type II error and attempted to avoid the term statistical power. The current simulations did not systematically create situations in which the null or alternative hypothesis of some of the selected tests was true or false. Rather, the groups differed in terms of the latent trait $c_l$. The hypotheses of the utilized tests were a priori unrelated to $c_l$. One may nevertheless consider a "type I error" simulations as presented by [70]. In these simulations, the groups are identical with respect to $c_l$ or some other latent parameter representing the empirical event. Crucially, the groups differ in terms of some other latent parameter. The main challenge with simulations with such scenario, is the choice and justification of this auxiliary parameter. While $c_l$ parameter was similar across wide range of distributions and a justification based on measurement theory was provided, the auxiliary parameters differed. [70] selected the latent scale parameter of OLRM. While for some CFE distributions a similar latent scale parameter is available and commonly used (e.g. $c_v$ of Log-normal and Logit-normal distribution), a theoretical work is required to derive such parametrization for other CFE distributions. Furthermore, researchers need to ask whether a case in which two groups differ in terms of latent scale but not in terms of latent mean represents a plausible empirical situation. As already mentioned, numerous robustness studies have been concerned with cases of equal means and heterogeneous group variances. [64] expressed the concern that such situation does not well represent the empirical reality. The current study further supports this view by suggesting that if the measure is affected by CFE, then a decrease in mean is associated with a decrease in variance through $c_l$ (or $c_u$). With a heterogeneous latent scale, it is somewhat easier to conceive of a relevant research situation in which such "type I error" may occur. For instance, one may imagine that some cognitive measure is affected by both CFE and intelligence. Then if the groups are not balanced in terms of participant's sex and assuming that intelligence of male population varies more than that of a female population (while the sexes do not differ in terms of mean intelligence), then the two groups will differ in terms of latent variance. Needless to say, this assumes that the latent parametrization accurately represents the empirical structure (in this case intelligence). Hence, the case of CFE-affected measure with groups equal in terms of $c_l$ and $c_u$, but heterogeneous in terms of some other parameter, may be of some interest, but requires further theoretical work which would substantiate the choice of the auxiliary latent parameter.

**Hierarchical structures.** In discussions of robustness of application of linear statistical methods to questionnaire data it is common to consider a situation with multiple items that measure the same latent trait. Then it is argued that averaging across items improves the robustness (e.g. [43], see [70] for a different opinion). We didn't consider hierarchical or other non-nested structures in the current work as we think that these introduce different type of difficulties that requires a separate investigation. To briefly sketch the difficulties, consider $Y_{ij}$ to be the response of participant $i$ on item $j$ of a questionnaire. Then for equation

$\text{mean}_i(\text{mean}_j(Y_{ij})) = \text{mean}_j(\text{mean}_i(Y_{ij}))$ to hold, it is necessary for variance of $Y_{ij}$ to be equal across $i$ and $j$. If CFE affects $Y_{ij}$ and its magnitude varies across items, then variance will vary across items and $\text{mean}_j$ will provide a biased estimate. A popular solution ([30] section 12.3) is to use weighted mean, where less noisy observations are given more weight, thereby increasing the overall accuracy of the estimator. If CFE results in a noisy inference, then such estimator will tend to ignore items with CFE. If however CFE results in biased noisy inference, then the weighting procedure will give more weight to items with CFE and will increase the overall influence of CFE on the mean. This example illustrates that CFE-affected measures applied in hierarchical and non-nested design require a separate investigation.

**Tobit minimum/maximum.** The current work focused exclusively on CFE defined by essential maximum/minimum rather than CFE defined by Tobit maximum/minimum. The aim was to fill in the gap left by the existing research on CFE. It is difficult to compare the results between the essential maximum CFE and Tobit CFE, since the Tobit research is mostly focused on comparison of linear regression with Tobit regression on test scores of human subjects. The general finding that Tobit regression provides better fit to data and different estimates than the linear methods echoes current results. Furthermore, difficulty in dealing with interactions (e.g. inaccurate estimation of regression coefficients of interaction terms [75]) was observed in Tobit research as well as in the current work. To discuss the difference between the two types of CFE, recall that we observed in section 4.2 that the probability distributions with essential minimum showed nonlinearity with weak CFE, which is in contrast with Tobit model, which shows linear behaviour when CFE is weak. As noted, it is possible to obtain a shift between linear and nonlinear regime with Gamma distribution [24]. The question of a suitable parametrization of Gamma distribution to obtain regime-switching behaviour and how the resulting distribution compares to Tobit model require further investigation. However, the case of Gamma distribution suggests, that the distinction between Tobit minimum and essential minimum is less pointed than it is presented in the literature [9, 13] and less sharp than it was presented in the literature overview of this report. We think, that it may be possible to formulate Tobit minimum as a limiting case of essential minimum. Then Tobit model may be obtained as a limiting case of some probability distribution with essential minimum/maximum.

**Choice of parametrization.** We feel obliged to stress again, without having much to add, that the presented results crucially depend on the choice of parametrization. This includes the choice of nuisance parameters. Comparisons of current results with those from other studies need to take the discrepancy in parametrization into account. The results in Fig 11 suggest that the parametrization was reasonable as in most cases the distributions satisfied the informal CFE conditions.

## 4.5 Recommendations

The performance comparison between statistical methods is perhaps of most importance to the applied researcher and we conclude the report on this topic. Use of data transformation has great potential with continuous measures, but requires a careful consideration of how CFE affects the measurement tool, otherwise it may be ineffective. With discrete data and with more complex study design, use of generalized linear models is appropriate. Again, the model assumptions must accurately match how CFE affects the measure. Rank-based tests performed fairly well when considered relative to other alternatives. The results however suggest that methods that utilize rank-transform perform similar to methods that utilize log or logit transform. As with other transforms, if the fit between the choice of rank-based transform and the data generating mechanism is poor, test performance degrades and main effects and

interactions may not be correctly identified. The same applies to linear methods. However, across wide range of scenarios, $t$-test and $F$-test showed inferior performance and their use with data with CFE should be discouraged. Perhaps, application of Welch's $t$-test may be defended on the grounds that, it "only" leads to a (possibly negligible) loss of power.

Thus, in order to make the proper choice, the researcher requires knowledge how the measurement tool numerically represents the concatenation operation and hence how CFE affects the measurement. In an ideal case, this information was obtained by validation and calibration studies done by the author/manufacturer of the measurement tool and is provided in the manual. If this information is not available, as is often the case, the researcher has to perform such studies herself. Note that such validation/calibration procedure is advisable when handling any kind of nonlinearity not just CFE. At minimum, the researcher should establish the numerical range of the measurement tool in which linear behaviour can be assumed and then she should restrict the conclusions to the corresponding range of the latent trait. Researchers wishing for more general inferences may consider whether the bounds of the numerical range create a CFE with Tobit or with essential maximum/minimum. In an ideal case, the researcher would obtain a parametric functional specification of ceiling and floor effect. Our review of formal measurement theory and POME provided the language to formulate and express such specification. We encourage researchers to use this formalism.

The use of modern inferential methods, that were considered in the current work, can't be recommended in their current form. The trimmed $t$-test showed worst performance of all tests and was ineffective at countering CFE. The equivalence testing methods, depended on the correct data transformation and otherwise produced false rejection of alternative hypothesis when the measure was affected by CFE. On occasion, confidence intervals manifested patterns of biased inference, where the estimate became more biased and more certain as the magnitude of CFE increased. Cohen's $d$ was biased by CFE as well and hence its use in meta-analysis or for research planning with measures affected by CFE is problematic. For research planning, we recommend the use of fake data simulation([30] chapter 8.1) which we illustrated with research scenario from [1].

Apart from robustness, computational costs and difficulty of interpretation may be considered as main hurdles to wider application of generalized linear models. With the recent advance of Monte Carlo methods, packages for fitting generalized linear models are readily available (e.g. [101]). Such software makes the fitting process fast and straightforward. If not already the case, computational cost of generalized linear models should cease to be a hurdle in the foreseeable future. Interpretation of GLMs is somewhat more involved, but for instance if CFE is modelled with the logarithm function, an additive change in the latent trait may be interpreted as a multiplicative change in the measured quantity, which can be expressed and interpreted as a percentage increase. Change in Logit function, may be interpreted as additive change (for values around 0.5) or as multiplicative change (for values near 0 and 1). [30] and other textbooks provide good guidance on the interpretation of estimates obtained with generalized linear models.

Finally, one may consider modification of the measurement tool itself with the goal of mitigating CFE. [106] and [107] discuss revision of questionnaire norms in the extremal regions which goes towards our appeal for further validation studies, rather than being a modification of a measurement tool itself. [71] proposed to omit the extremal data from analysis. Given the performance of trimmed $t$-test in the current work, we doubt that such omission can effectively counter CFE. [108] proposed to expand the options of a rating scale, which may perhaps mitigate, but will not remove CFE. If one considers that any data transformation or even model fitting procedure can be made evaluative part of the measurement tool then, what has been written about the utility of transformations and about the necessity of validation/calibration

studies, applies to measure design as well. If the evaluative procedure of the measurement tool is designed poorly, this may lead to a loss of information (unless the researcher has an access to validation/calibration data). While successful data analysis may turn biased inference or biased noisy inference into noisy inference, correct inference can't be obtained with data-analytic solution and a modification of measurement tool is the only option to avoid or at least mitigate noisy inference.

Irrespective of the recommendations, we hope, that the current work will alert researchers to the types of detrimental effects associated with CFE and that it will motivate the design of new effective statistical solutions and new measurement tools, that avoid the bias and noise created by CFE.

## Supporting information

**S1 Appendix. Figures with results of supplementary analyses.** See the introductory paragraph of section 3 for details.
(PDF)

## Author Contributions

**Conceptualization:** Matúš Šimkovic, Birgit Träuble.

**Data curation:** Matúš Šimkovic.

**Formal analysis:** Matúš Šimkovic.

**Funding acquisition:** Birgit Träuble.

**Methodology:** Matúš Šimkovic.

**Project administration:** Birgit Träuble.

**Software:** Matúš Šimkovic.

**Supervision:** Birgit Träuble.

**Visualization:** Matúš Šimkovic.

**Writing – original draft:** Matúš Šimkovic, Birgit Träuble.

**Writing – review & editing:** Matúš Šimkovic, Birgit Träuble.

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
