## [Decision Letter · Decision Letter 0]

26 Jul 2019

Robustness of statistical methods when measure is affected by ceiling and/or floor effect

PONE-D-19-14718

Dear Dr. Simkovic,

We are pleased to inform you that your manuscript has been judged scientifically suitable for publication and will be formally accepted for publication once it complies with all outstanding technical requirements.

With kind regards,

Alan D Hutson

Academic Editor

PLOS ONE

Additional Editor Comments (optional):

Please attend to the typos noted by one reviewer.

Reviewers' comments:

Reviewer's Responses to Questions

**Comments to the Author**

1. Is the manuscript technically sound, and do the data support the conclusions?

Reviewer #1: Yes

Reviewer #2: Yes

2. Has the statistical analysis been performed appropriately and rigorously? 

Reviewer #1: Yes

Reviewer #2: Yes

3. Have the authors made all data underlying the findings in their manuscript fully available?

Reviewer #1: Yes

Reviewer #2: Yes

4. Is the manuscript presented in an intelligible fashion and written in standard English?

Reviewer #1: Yes

Reviewer #2: Yes

5. Review Comments to the Author

Reviewer #1: If possible, as written in the attached review report, application of proposed methods on real data sets may be shown.

Reviewer #2: 1. This study investigates the effects of ceiling and/or floor effects on hypothesis testing, such as t-test, F-test, in addition, on estimations of effect sizes and confidence interval, etc. The author presented explicit results and concluded that the presence of CFE results in bias, divergence in estimation and testing after comparing different statistical methods with different data distributions. The uniqueness in this study is the use of measurement theory in defining CFE and using this theory in data generation. This study contributes to the CFE research and is worthy of publication.

2. There are minor typos in the manuscript, please read throughly and correct them , e.g., Line 511, Line 958…

6. PLOS authors have the option to publish the peer review history of their article (what does this mean?). If published, this will include your full peer review and any attached files.

Reviewer #1: No

Reviewer #2: No

---

## [Editor Report · Acceptance letter]

2 Aug 2019

PONE-D-19-14718

Robustness of statistical methods when measure is affected by ceiling and/or floor effect 

Dear Dr. Šimkovic:

I am pleased to inform you that your manuscript has been deemed suitable for publication in PLOS ONE. Congratulations! Your manuscript is now with our production department. 

With kind regards,

on behalf of

Dr. Alan D Hutson 

Academic Editor

PLOS ONE